# Linalool combats *Saprolegnia parasitica* infections through direct killing of microbes and modulation of host immune system

Tao Tang[1], Weiming Zhong[1], Puyu Tang[2], Rongsi Dai[1], Jiajing Guo[3]*, Zhipeng Gao[1]*

[1]Hunan Engineering Technology Research Center of Featured Aquatic Resources Utilization, Fisheries College, Hunan Agricultural University, Changsha, China; [2]State Key Laboratory of Freshwater Ecology and Biotechnology, Institute of Hydrobiology, Chinese Academy of Sciences, Wuhan, China; [3]Hunan Agriculture Product Processing Institute, Hunan Academy of Agricultural Sciences, Changsha, China

## eLife Assessment

This **useful** study presents findings on the efficacy and mechanisms of linalool protection against Saprolegnia parasitica oomycetes in the grass carp model. The evidence presented is **solid** since the methods, data and analyses broadly support the claims with only minor weaknesses. This work will be of great interest to scientists within the fields of aquaculture, ichthyology, microbiology, and drug discovery.

*For correspondence:
guojiajing1986@163.com (JG);
gaozhipeng627@163.com (ZG)

Competing interest: The authors declare that no competing interests exist.

**Abstract** *Saprolegnia parasitica* is one of the most virulent oomycete species in freshwater aquatic environments, causing severe saprolegniasis and leading to significant economic losses in the aquaculture industry. Thus far, the prevention and control of saprolegniasis face a shortage of medications. Linalool, a natural antibiotic alternative found in various essential oils, exhibits promising antimicrobial activity against a wide range of pathogens. In this study, the specific role of linalool in protecting *S. parasitica* infection at both in vitro and in vivo levels was investigated. Linalool showed multifaceted anti-oomycetes potential by both of antimicrobial efficacy and immunomodulatory efficacy. For in vitro test, linalool exhibited strong anti-oomycetes activity and mode of action included: (1) Linalool disrupted the cell membrane of the mycelium, causing the intracellular components leak out; (2) Linalool prohibited ribosome function, thereby inhibiting protein synthesis and ultimately affecting mycelium growth. Surprisingly, meanwhile we found the potential immune protective mechanism of linalool in the in vivo test: (1) Linalool enhanced the complement and coagulation system which in turn activated host immune defense and lysate *S. parasitica* cells; (2) Linalool promoted wound healing, tissue repair, and phagocytosis to cope with *S. parasitica* infection; (3) Linalool positively modulated the immune response by increasing the abundance of beneficial Actinobacteriota; (4) Linalool stimulated the production of inflammatory cytokines and chemokines to lyse *S. parasitica* cells. In all, our findings showed that linalool possessed multifaceted anti-oomycetes potential which would be a promising natural antibiotic alternative to cope with *S. parasitica* infection in the aquaculture industry.

## Introduction

Saprolegniasis is one of the most serious oomycete diseases in the aquaculture industry all over the world, infecting a wide host range (fish, crustaceans, amphibians, and their eggs) with a conspicuous symptom (grayish-white cotton wool-like covering on the surface of host) (*Derevnina et al., 2016*; *Tandel et al., 2021*). Oomycete infections lead to significant economic losses and serious damage to natural ecosystems (*Derevnina et al., 2016*). As the infection intensifies, saprolegniasis can cause muscle ulceration, compromised immune system, respiratory problems, secondary infection, and finally result in host death (*Wang et al., 2021b*). *Saprolegnia*, *Achlya*, and *Leptolegnia* are key pathogens responsible for saprolegniasis outbreaks (*Costa and Lopes, 2022*), among them, *Saprolegnia* is proven to be the most virulent (*Sandoval-Sierra et al., 2014*; *Khulbe et al., 1995*). *S. parasitica*, *S. ferax*, *S. delica*, and *S. diclina* are the main pathogens causing saprolegniasis (*Engblom et al., 2023*), as a highly virulent group belonging to the phylum Oomycota, *S. parasitica* serves as the primary species in fish infections (*Money et al., 2004*).

In the past, malachite green was highly effective in controlling *Saprolegnia* infections, however, it has been banned worldwide for its carcinogenic and toxicological effects, which led to a sharp re-emergence of *Saprolegnia* infections in aquaculture industry (*van West, 2006*). Recently, various methods have been established to deal with *S. parasitica* infections, mainly including physical (water quality management, temperature control, and so on), chemical (hydrogen peroxide, natural antimicrobial compounds, and so on), and biological (probiotics, antagonistic microbe, and so on) methods (*van den Berg et al., 2013*). Among these, antibiotics have served as the primary therapeutic and prophylactic tools for saprolegniasis management in aquaculture. However, their overuse and abuse have resulted in antibiotic resistance among various aquatic pathogens (*Yu et al., 2023*). Therefore, the increasing need for sustainable and eco-friendly strategies to combat *Saprolegnia* infections has driven research into natural antimicrobial agents, due to their advantages in terms of avoiding antibiotic resistance issues, health safety, and ecological sustainability (*Gao et al., 2023b*).

Plant extracts regulate specific and non-specific immunity, promoting immune responses and playing a crucial role in improving production performance, enhancing antioxidant capacity, combating viruses, preventing diseases, and improving gut health (*Liu et al., 2018*). This immunomodulatory function is mainly attributed to the secondary metabolites found in plants, such as flavonoids, polysaccharides, lactones, alkaloids, diterpenoids, and glycosides. These secondary metabolites can not only directly enhance immune capacity but also balance the gut microbiota and strengthen the intestinal mucosal barrier. After ingestion, plant extracts improve the immune defense mechanisms of intestinal epithelial cells, resist diseases caused by pathogenic microorganisms, and effectively enhance the body's resistance to pathogen infections (*Shehata et al., 2022*). In our earlier studies, the anti-oomycetes activities of 12 plant essential oils (EOs) and 5 of their major compounds (linalool, pinene, limonene, myrcene, and terpinene) were evaluated, and we found that linalool was one of the strongest anti-oomycetes compounds and selected for further study (*Tang et al., 2024*). Linalool, a monoterpene alcohol found in various EOs, has shown promising antimicrobial activity against a wide range of pathogens (*Peana et al., 2002*). Our groups and other research teams have reported that linalool exhibits antimicrobial activity against bacteria such as *Escherichia coli* (*Gao et al., 2023b*), *Listeria monocytogenes* (*Gao et al., 2021*), *Elizabethkingia miricola*, *Streptococcus pyogenes* (*Praseetha et al., 2023*), *Pseudomonas fragi* (*Li et al., 2022a*), *Staphylococcus aureus* (*Aytac et al., 2017*), *Brochothrix thermosphacta* (*He et al., 2022*), and *Aeromonas hydrophila* (*Zhong et al., 2021*), as well as against oomycetes like *Colletotrichum lagenarium* (*Jiang et al., 2023*) and *Phytophthora capsici* (*Yang et al., 2022*), and fungi including *Aspergillus flavus* (*Li et al., 2022b*), *Penicillium citrinum*, *Chaetomium globosum* (*Satoh et al., 2022*), *Colletotrichum gloesporioides*, *Fusarium oxysporum* (*Barros Gomes et al., 2019*), *Botrytis cinerea* (*Shen et al., 2022*), and *Candida albicans* (*Marcos-Arias et al., 2011*).

In our previous study, we found that linalool exhibited strong antimicrobial activity against *S. ferax* (*Tang et al., 2024*), while whether it is also a promising agent against *S. parasitica*, another key virulent pathogen for *Saprolegniasis* is still unknown, and the detailed mechanism, especially the in vivo immune mechanism induced by linalool remains poorly understood. Thus, in this study we focused on *S. parasitica* and aimed to investigate the specific role of linalool in protecting *S. parasitica* infection at both of in vitro and in vivo level by using different technologies, including in situ fluorescence observation, microstructure observation, molecular docking, AlphaFold2, transcriptome profiling,

histological examinations, gut microbiota analysis, and deep bioinformatics analysis. We believe this study will shed light on the antimicrobial activity and mode of action of linalool against *S. parasitica*, and contribute to the development of novel antibiotic alternative strategies to cope with *S. parasitica* infection in aquaculture industry.

## Results

### The in vitro anti-oomycete activity of linalool against *S. parasitica* CQT2

*Figure 1A* illustrates the anti-oomycetes activities of EOs and their major components against *S. parasitica* CQT2. Notably, linalool and the EO extracted from daidai flowers were particularly effective in restraining mycelium growth.

After adding 5 μl of resazurin dye, when the color of the wells turned to pink, the linalool concentration in the first non-pink well corresponded to the MIC. The culture liquid in the hole where no mycelium growth was seen was marked onto the plate and incubated at 25°C for 7 days. The well with the lowest linalool concentration and no mycelium growth was identified as minimum fungicidal concentration (MFC). Linalool exhibited the MIC and MFC of 0.025% and 0.025% (vol/vol, 0.216 mg/ml) against spores, respectively (*Figure 1B*). Besides, both the MIC and MFC against mycelium growth were 0.05% (vol/vol, 0.432 mg/ml) and 0.1% (vol/vol, 0.864 mg/ml), respectively. Various linalool doses were examined against *S. parasitica* CQT2 on linalool-containing plates to evaluate their impact on inhibiting mycelium growth. As shown in *Figure 1C, D*, the findings indicated the anti-oomycetes activity of linalool against *S. parasitica* CQT2 was dose dependent. *Figure 1E* displays mycelium growth on potato dextrose agar (PDA) plates under varying linalool concentrations. The viability of mycelium following linalool treatment was assessed using a prodium iodide (PI) staining assay, where non-viable (dead) mycelium appeared red after PI uptake. Untreated mycelium showed no detectable red fluorescent, indicating 100% viability. In contrast, distinct red fluorescent mycelium was observed after linalool treatment at both MIC and 2× MIC levels (*Figure 1F*).

### Effects of linalool on cell membrane damage of *S. parasitica* CQT2

Confocal microscopy using amphiphilic styryl dye FM4-64 has been employed to study endocytosis and vesicle trafficking in living fungal hyphae, while SYTO 9, a fluorescent nucleic acid stain, is widely used in microbiology, particularly in fluorescence microscopy, to evaluate bacterial viability (*Fischer-Parton et al., 2000*). The in situ fluorescence observations are displayed in *Figure 2A*, with linalool treatment for 0 min, the mycelium structure remained undamaged, and the red and green fluorescence showed an even distribution. After 2 min, red fluorescence started appearing within the mycelium, accompanied by the entry of the styryl dye FM4-64, indicating damage to the cell membrane. As the duration of linalool treatment increased, the intensity of red fluorescence within the mycelium also increased, indicating a rise in the number of mycelial cell deaths along with severe damage to the cell membrane. When examined by scanning electron microscopy (SEM), the mycelium exhibited a smooth and intact surface in the control group (*Figure 2B*), while in the treated groups (*Figure 2C*), it appeared wrinkled, deformed, and shrunken (red dashed quadrilateral). When examined by transmission electron microscopy (TEM), the untreated mycelium cell exhibited clear and intact ultrastructure in the cell membrane, cell wall, mitochondria, and rough endoplasmic reticulum (*Figure 2D*). However, after treatment with linalool, the cell membrane was wrinkled and seriously damaged, and mitochondria began to dissolve (*Figure 2E*). As shown in *Figure 2F, G*, the cell membrane structure disappeared, the cell wall became thinner, and some exhibited severe deformation and bending. Vacuoles appeared in the cytoplasm, and mitochondria and rough endoplasmic reticulum ruptured and dissolved (red circles). No intact organelles were present within the cytoplasm.

As shown in the Gene Ontology (GO) classification analysis (*Figure 2H*), the cellular components were distributed across eight groups, with 'cell part' and 'membrane part' being the most prevalent. The differentially expressed genes (DEGs) associated with the intrinsic component of the membrane were identified and categorized into four classes: linked to ribosome function, involved in DNA/RNA function, connected to signal transduction and regulation, and pertaining to cell metabolism (*Figure 2I*). GO classification analysis revealed a significant number of DEGs associated with the intrinsic components of the membrane. This finding is further supported by observations from SEM

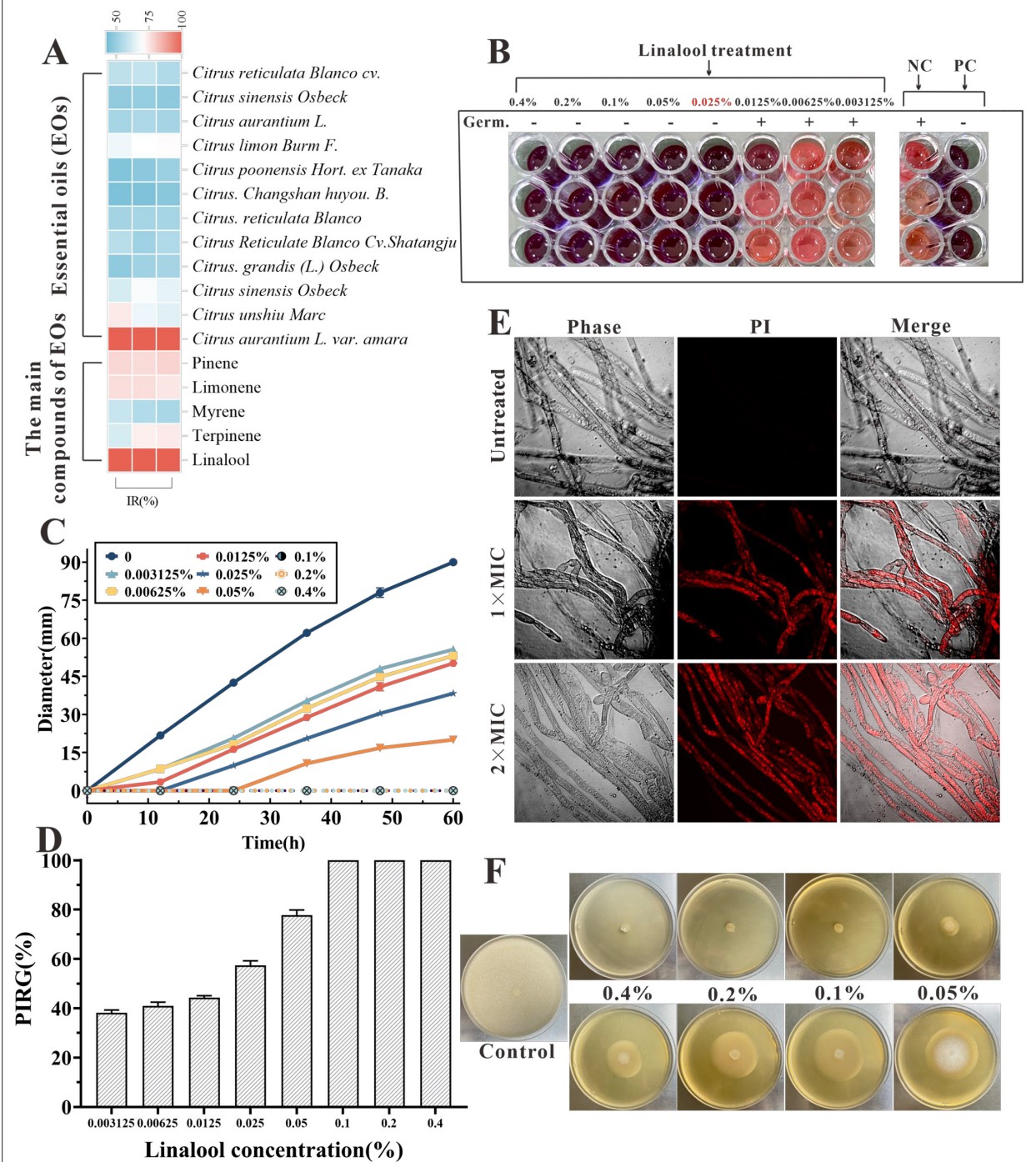

**Figure 1.** The in vitro anti-oomycetes activity of linalool against *S. parasitica* CQT2. (**A**) Inhibition rate (IR) of essential oils (EOs) and their major components against *S. parasitica* CQT2. (**B**) Determination of minimum inhibitory concentration (MIC) of linalool against *S. parasitica* spores, employing malachite green (2.5 mg/ml) as a positive control (PC) and Tween 20 as a negative control (NC). (**C**) Effects of linalool on mycelium growth inhibition over a 60-hr period at varying concentrations. (**D**) Examination of the percentage inhibition of radial growth (PIRG %) in mycelium treated with linalool after 60 hr. (**E**) Effects of linalool on the viability of *S. parasitica* CQT2 through a prodium iodide (PI) staining assay was conducted under three conditions: Control (no linalool), 1× MIC (0.05%) linalool treatment, and 2× MIC (0.1%) linalool treatment. (**F**) Effects of different concentrations of linalool on mycelium growth on potato dextrose agar (PDA) plates.

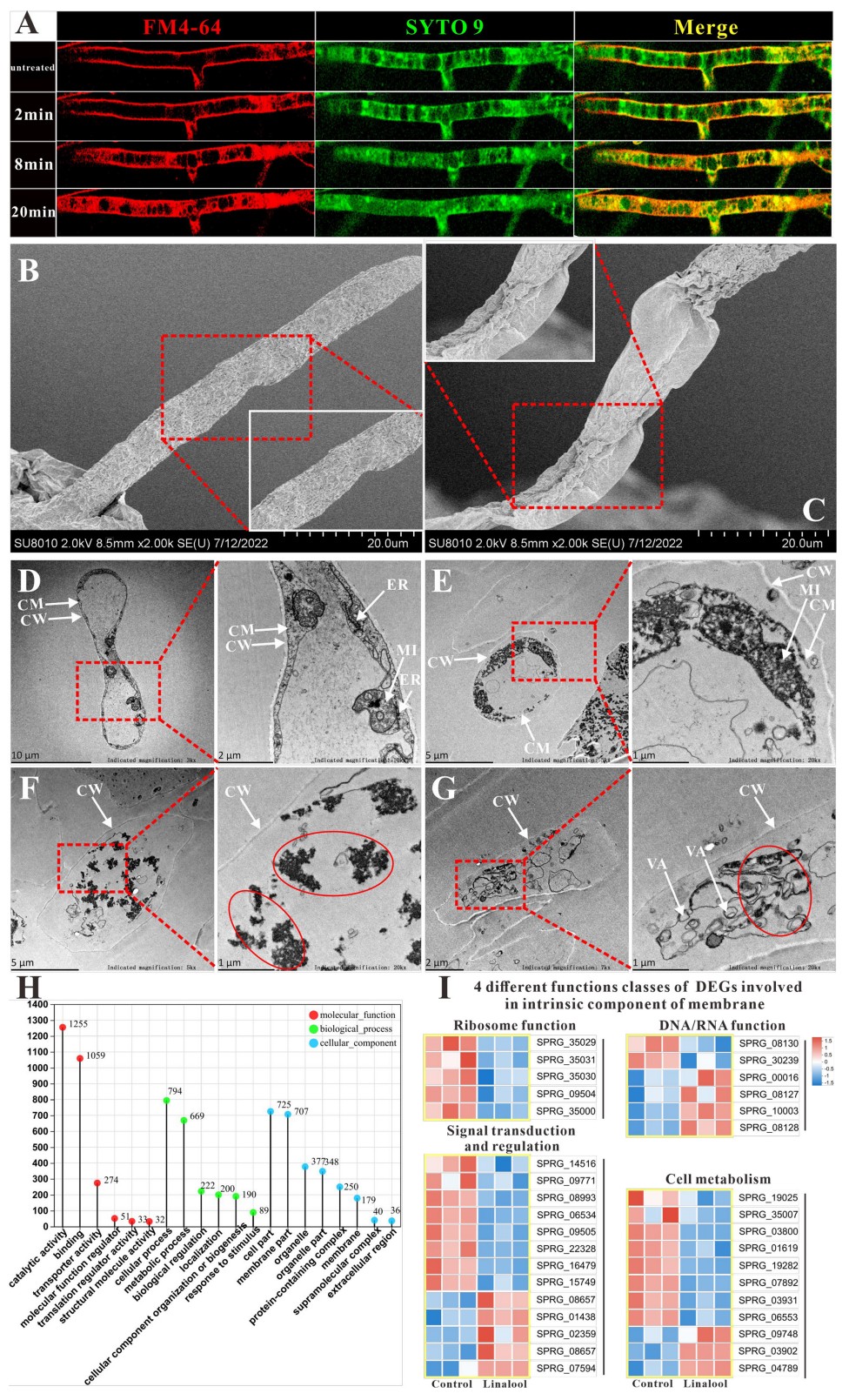

**Figure 2.** Effects of linalool on the cell membrane integrity of *S. parasitica* CQT2. (**A**) Observation of the effect of linalool at 6× minimum inhibitory concentration (MIC) (0.3%) on the cell membrane of *S. parasitica* CQT2 using the FM4-64 and SYTO 9 staining assay. Scanning electron microscopy (SEM) images of *S. parasitica* CQT2 mycelium without (no linalool, **B**) and with 1× MIC (0.05%, **C**) linalool treatment. Transmission electron microscopy (TEM)

*Figure 2 continued on next page*

*Figure 2 continued*

images of *S. parasitica* CQT2 mycelium without (no linalool, **D**) and with 1× MIC (0.05%, **E–G**) linalool treatment. CM: cell membrane; CW: cell wall; ER: rough endoplasmic reticulum; MI: mitochondria; VA: vacuoles. (**H**) The Gene Ontology (GO) classification of differentially expressed genes (DEGs). (**I**) Four different functions classes (linked to ribosome function, involved in DNA/RNA function, connected to signal transduction and regulation, and pertaining to cell metabolism) of DEGs involved in intrinsic component of membrane.

and TEM, which provide consistent evidence aligning with the GO classification analysis results. Therefore, the cell membrane may be a potential target for linalool.

## The in vitro anti-oomycete mechanisms revealed by transcriptome analysis

As shown in *Figure 3—figure supplement 1A, B*, a total of 4142 DEGs were discovered between the control and treatment groups, with 2016 showing upregulation and 2108 downregulation. *Figure 2H* presents the GO analysis, providing a comprehensive evaluation of the functional aspects of genes and their products. These DEGs were categorized into 20 GO categories, with 6 associated with biological activities, where cellular and metabolic functions had the most DEGs. There were also six molecular function categories, with catalytic activity and binding containing the majority of DEGs. Additionally, *Figure 3—figure supplement 1C* demonstrates the results of Kyoto Encyclopedia of Genes and Genomes (KEGG) annotation, revealing 10 metabolic (M) pathways, 2 environmental information processing pathways, 2 cellular processes, and 5 genetic information processing.

### The ribosome biogenesis and protein synthesis of *S. parasitica* were prohibited by linalool

*Figure 3A, B* displays the results of GO enrichment analysis, revealing the top 20 biological processes that differed significantly between the control and treated groups. Notably, in *Figure 3B*, the downregulated biological processes are highlighted, and remarkably, 19 of them were associated with 'ribosome', such as ribosome biogenesis, ribonucleoprotein complex biogenesis, pre ribosome, and small-subunit process some. In addition, KEGG enrichment analysis in *Figure 3C* revealed that 'Ribosome biogenesis in eukaryotes' was the most enriched pathway, aligning with the GO enrichment analysis results mentioned earlier. The pathway of ribosome biogenesis in eukaryotes (*Figure 3D*) showed that linalool influenced DNA transcription, tRNA transport, rRNA processing and maturation (5.8S, 18S, and 25S), and the biogenesis and assembly of ribosome subunits (40S and 60S) in the cell, which might lead to the reduction of *S. parasitica* growth. We found that many DEGs related to ribosome biogenesis (*Figure 3D*) and RNA polymerase (*Figure 3E*) are downexpressed. *Figure 3A* highlights the upregulated biological processes, and notably, 11 of them were linked to 'amino acids', such as the sulfur amino acid, L-phenylalanine, alpha-amino acid, tyrosine, and homocysteine. Amino acids serve essential roles within the cell, acting as both the foundational components for synthesizing new proteins and as precursors for various metabolic processes (*Yan and Lamb, 2012*).

The results of GO enrichment analysis and KEGG enrichment analysis suggested that linalool might interact with specific domains of the ribosome, potentially affecting its structure and function. To determine whether linalool could bind to ribosome-associated upstream proteins in *S. parasitica*, molecular docking was conducted to explore the specific binding proteins of linalool with the ribosome. The binding energy of linalool was –6.2 kcal/mol with NOP, –4.8 kcal/mol with SNU13, and –5.0 kcal/mol with DKC1 (Table S1). Consequently, NOP1 was selected as the receptor protein for subsequent molecular docking. *Figure 3F* illustrates the tertiary structure of ribosomal protein NOP1. Linalool formed hydrogen bonds with glutamic acid (Glu155) and phenylalanine (Phe156) of NOP1 while exhibiting hydrophobic interactions with alanine (Ala181), arginine (Arg182), valine (Val201), and isoleucine (Ile209) (*Figure 3G*). In summary, this interference may hinder the ribosome's ability to execute normal biological processes, such as protein synthesis, thereby impacting the growth and reproduction of oomycetes.

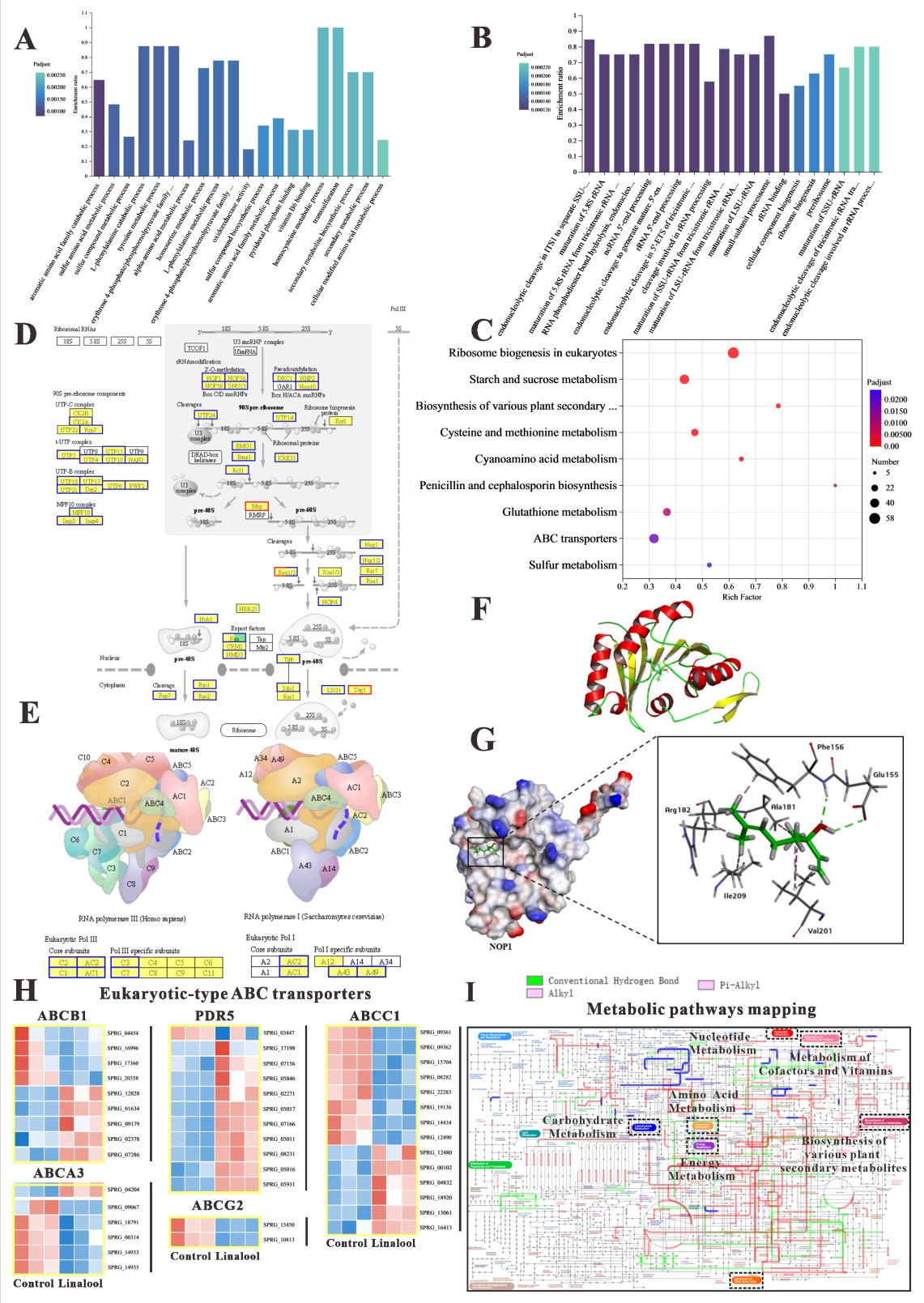

**Figure 3.** The in vitro anti-oomycete mechanisms revealed by transcriptome analysis. (**A, B**) The Gene Ontology (GO) enrichment of up- and downregulated differentially expressed genes (DEGs). (**C**) The Kyoto Encyclopedia of Genes and Genomes (KEGG) enrichment of DEGs. (**D**) Comparison of ribosome biogenesis in eukaryotes pathway between the linalool treated mycelium and the control group. (**E**) Comparison of RNA polymerase pathway between the linalool treated mycelium and the control group. (**F**) The tertiary structure of NOP1 was displayed using a cartoon representation.

*Figure 3 continued on next page*

Figure 3 continued

(**G**) Molecular docking of linalool with NOP1 was performed by enlarging the regions binding to the NOP1 activation pocket to showcase the detailed amino acid structures, which were presented using a surface model, while the small molecule was displayed with a ball-and-stick representation.
(**H**) Comparison of ABC transporters pathway between the linalool treated mycelium and the control group. The red squares represented upregulated genes, and the blue squares represented downregulated genes. (**I**) Visual analysis of metabolic pathways with iPath3.0. The figure represented gene set annotated pathways, red and green represented pathways annotated by genes in different gene sets, respectively, and blue represented pathways co-annotated by genes in two gene sets.

The online version of this article includes the following figure supplement(s) for figure 3:

**Figure supplement 1.** Global transcriptomic analysis between the treated (mycelium treated with linalool) and control group (untreated with linalool).

**Figure supplement 2.** Global transcriptomic analysis after *S. parasitica* infection and linalool treatment in grass carp.

## The global metabolisms of *S. parasitica* were regulated by linalool

In KEGG enrichment analysis, we found that six processes were associated with metabolic pathways, such as cysteine and methionine metabolism, starch and sucrose metabolism, and sulfur metabolism. Moreover, metabolic pathways were the most annotated pathway which was in line with the results of KEGG annotation analysis. As many changes were linked to metabolism, the Interactive Pathways Explorer (iPath) was employed to comprehend the global differential metabolic response in *S. parasitica* CQT2 (**Figure 3I**) which identified that linalool affects the global metabolic regulation of *S. parasitica* CQT2.

## ABC transporters of *S. parasitica* were affected by linalool

Some other important pathways were significantly enriched in our study, such as ABC transporter and sulfur relay system. The DEGs involved in ABC transporters (ABCA3, ABCB1, ABCC1, ABCG2, and PDR5) are shown in **Figure 3H**. In brief, these results demonstrated that cell membrane, protein synthesis, functions related to metabolism, and ribosome were significantly impacted by linalool and were potential drug targets of linalool.

## The in vivo protective effect of linalool against *Saprolegniosis*

To study the protective effects of linalool on *S. parasitica* infection in fish, linalool was added to grass carp after *S. parasitica* infection (linalool therapeutic group, LT) or 48 hr before infection (linalool prophylactic group, LP), and a positive control group (PC) without linalool and a negative control group (NC) without infection and linalool were set up (**Figure 4A**). The results showed that the survival rate of *S. parasitica* infected grass carp was significantly improved after treatment with linalool (**Figure 4C**), which proved greatly anti-*Saprolegnia* activity of linalool. In **Figure 4B**, there was no observable mycelium in both LT and LP groups, while the PC group displayed significant mycelium growth along with inflammation on the dorsum. The LZM activity was significantly increased in LP and LT group on the seventh day (**Figure 4E**), yet the alkaline phosphatase (AKP) activity (**Figure 4D**) was reduced. However, there was no significant difference in the superoxide dismutase (SOD) activity (**Figure 4F**) between LP and LT groups during the whole feeding trial.

To explore the histomorphology alterations caused by *S. parasitica* infection and linalool treatment, we performed histomorphology evaluations through HE staining (**Figure 4G**). The findings revealed significant pathological changes in the kidney, liver, spleen, and skin of grass carp due to *S. parasitica* infection. In PC group, severe kidney damage was evident, characterized by enlarged glomeruli, glomerular cysts, interstitial bleeding, and infiltration of inflammatory cells. In addition, extensive vacuolation and expansion of intercellular space were observed in hepatocytes, with nuclei displaced toward one side. Cytoplasmic pyknosis and structural disintegration were observed in the pancreatic acinar cells. Meanwhile, the red and white pulp of the spleen was poorly demarcated, the splenic corpuscle was irregularly shaped and arranged, and the white pulp was structurally disrupted. Furthermore, there was a notable increase in the number of macrophages and an expansion in the size of melanoma–macrophage centers. *S. parasitica* not only caused damage to both the dermal and epidermal layers but also induced muscle cell degeneration. Continuous treatment of linalool for 7 days significantly improved kidney, liver, spleen, and skin morphological integrity.

In the LP and LT groups, linalool alleviated tissue damage caused by *S. parasitica* infection on the dorsal surface of grass carp, enhancing the healing capacity (**Figure 4B**). Histomorphology

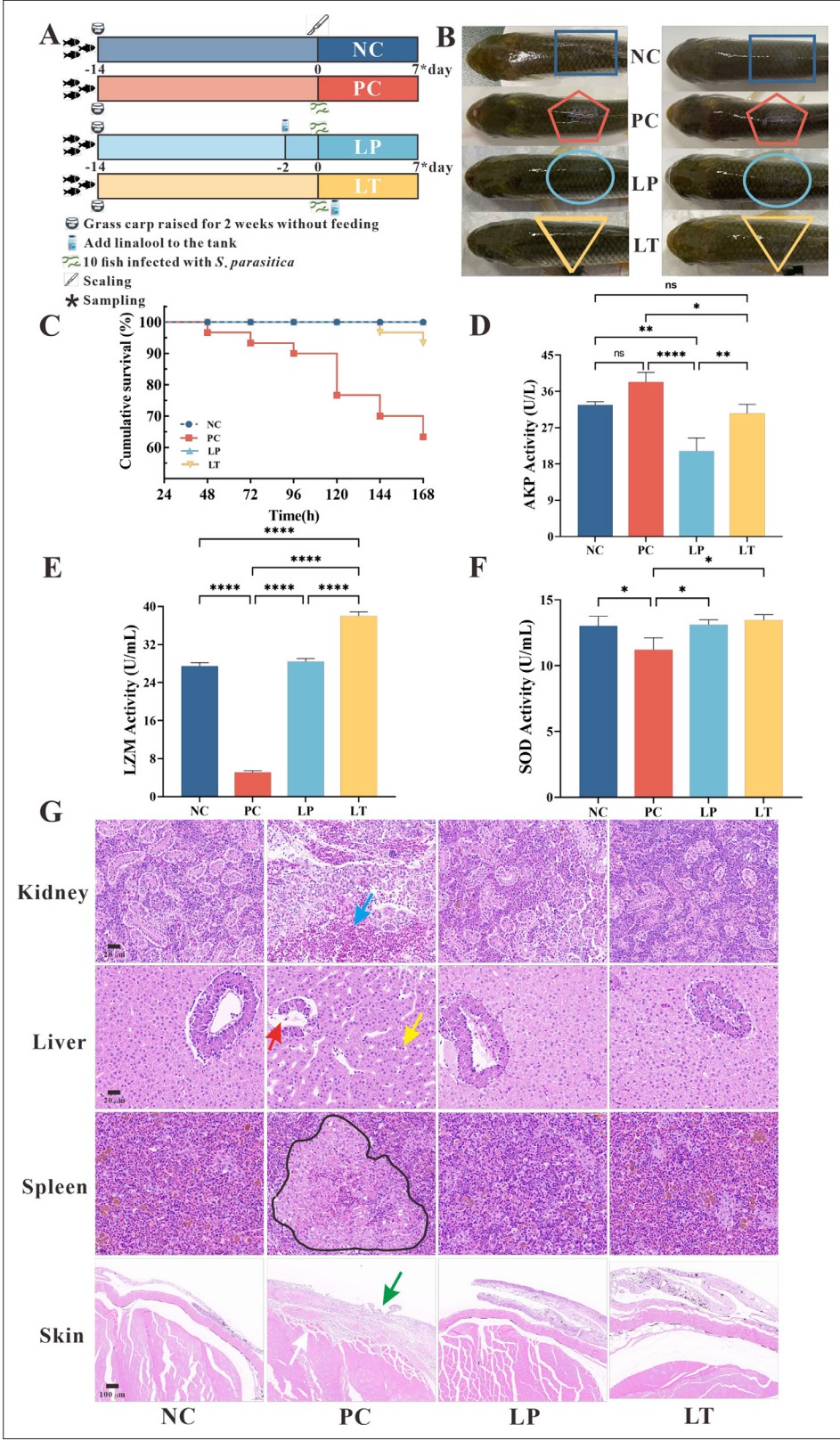

**Figure 4.** Protective effect of linalool on grass carp infected with *S. parasitica*. (**A**) The experimental design. Grass carp were raised for 2 weeks without feeding, fish without infection and linalool (negative control group, NC), fish infected with *S. parasitica* (positive control group, PC), and 10 fish uninfected soaked water containing linalool for 2 days and then 1 × 10⁶ spores/ml secondary zoospores were added (linalool prophylactic group, LP),

*Figure 4 continued on next page*

*Figure 4 continued*

and fish infected with *S. parasitica* soaked for 7 days in water containing linalool (linalool therapeutic group, LT). (**B**) The symptoms of *S. parasitica* infection in grass carps of different groups. (**C**) The survival rates of grass carp infected with *S. parasitica* of different groups. (**D–F**) Alkaline phosphatase (AKP), lysozyme (LYZ), and superoxide dismutase (SOD) activities in serum of grass carp of different groups. The results were presented as the means ± SD and were analyzed using one-way ANOVA followed by post hoc tests (*p < 0.05, **p < 0.01, ****p < 0.0001). (**G**) Histopathological analysis grass carp tissues in different groups. The arrows of different colors indicated: inflammatory cell infiltration in the kidney (blue arrow), cytoplasmic pyknosis (red arrow), nuclei displaced toward one side (yellow arrow), the red and white pulp was poorly demarcated, and a larger volume of melano-macrophage centers (black area), critical damage to the epithelium (green arrow) and myofiber (white arrow).

demonstrated that linalool could improve the integrity of organ morphology in grass carp (*Figure 4G*). Transcriptomics of the spleen revealed that linalool promoted wound healing, tissue repair, and phagocytosis to cope with *S. parasitica* infection (*Figure 5E*). In all, our results indicated that linalool promoted wound healing in grass carp.

## The in vivo saprolegniasis protective mechanism of linalool revealed by comparative transcriptome analysis

To analyze the contrasting gene-level responses in grass carp spleens, a vital immune organ, following *S. parasitica* infection and linalool treatment, spleen samples were collected for transcriptome analysis. Compared to PC group, 2029 genes in LT group exhibited significant difference, with 809 DEGs upregulated and 1220 downregulated. Meanwhile, 671 genes were identified as significantly different, with 609 DEGs upregulated and 618 downregulated in LP group relative to PC group (*Figure 3—figure supplement 2A, B*).

Based on the GO annotation analysis, a total of 2219 unigenes in the LT group and 520 unigenes in the LP group were classified into 3 primary functional categories (*Figure 3—figure supplement 2C*). Within the biological process category, 'cellular process', 'metabolic process', and 'biological regulation' were the most prominent subcategories. In terms of cellular components, 'membrane part' and 'cell part' were highly represented. 'Binding' and 'catalytic activity' were the most prevalent subcategories within molecular functions. Regarding KEGG annotations, the largest cluster within organismal systems was the immune system, encompassing 209 genes (42.47%) in the LT group (*Figure 3—figure supplement 2D*) and 38 genes (50.00%) in the LP group (*Figure 3—figure supplement 2E*). In the environmental information processing category, most unigenes were associated with signal transduction, with 227 (65.60%) in the LT group and 38 genes (52.00%) in the LP group. These results suggested that the immune response and signal transduction pathways were crucial for grass carp in dealing with *S. parasitica* infections.

The GO enrichment analysis of the spleens illustrated in *Figure 5A, B* to reveal distinct biological process. Those downregulated biological processes are shown in *Figure 5A*, compared to PC group, the 'positive regulation of immune system process' term (GO:0002684) and 'regulation of body fluid levels' term (GO:0050878) were enriched with the greatest number of DEGs in LT group. It is worth noting that there are nine terms related to 'body fluids (especially blood)', such as regulation of coagulation (GO:0050818), blood coagulation (GO:0007596), and regulation of hemostasis (GO:1900046). In addition, those upregulated biological process, the 'ion binding' term (GO:0043167) was enriched with the greatest number of DEGs in LT group. *Figure 5B* illustrates the GO terms with the largest number of downregulated DEGs in LP group was 'mall molecule binding' (GO:0036094).

KEGG enrichment analysis was used to explore the functional pathways involving DEGs. The 20 significantly enriched up- and downregulated pathways of each group are shown in *Figure 5C, D*. Multiple pathways associated with immune response were enriched in downregulated genes after linalool treatment in LT group, including complement and coagulation cascades, Toll-like receptor signaling pathway, IL-17 signaling pathway, C-type lectin receptor signaling pathway, T cell receptor signaling pathway, B cell receptor signaling pathway, and so on (*Figure 5C*). Toll-like receptor signaling pathway, RIG-I-like receptor signaling pathway, complement and coagulation cascades, and chemokine signaling pathway were enriched with mostly upregulated genes. Furthermore, metabolic pathways were enriched with mostly downregulated genes in LP group. Some pathways associated with immune response were significantly enriched, such as Antigen processing and presentation, Fc

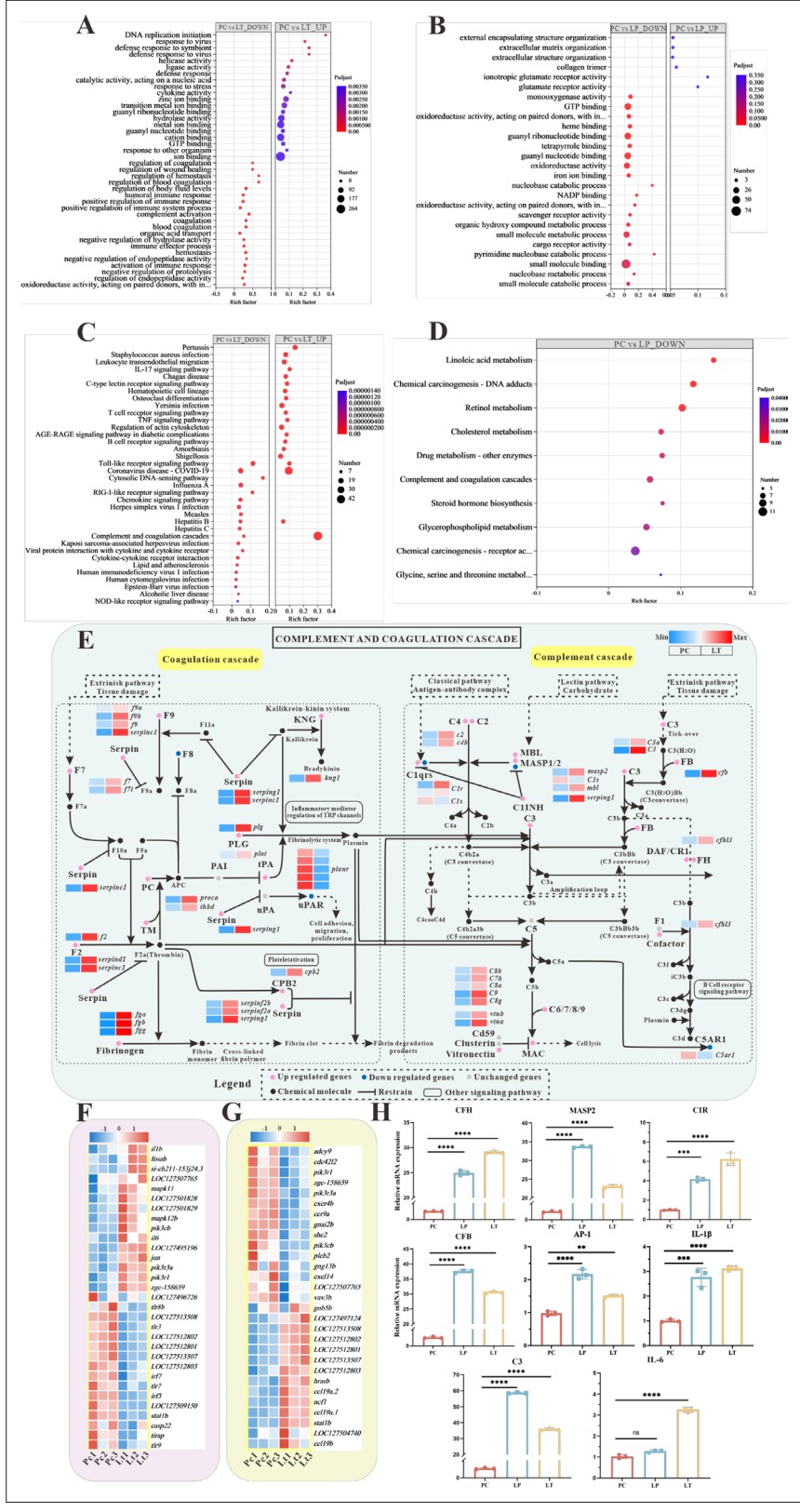

**Figure 5.** Global transcriptomic analysis after *S. parasitica* infection and linalool treatment in grass carp and in-depth analysis of crucial Kyoto Encyclopedia of Genes and Genomes (KEGG) pathway and differentially expressed genes (DEGs). (**A, B**) The Gene Ontology (GO) enrichment of up- and downregulated DEGs in linalool therapeutic (LT) and linalool prophylactic (LP) groups. (**C, D**) The KEGG enrichment of up- and downregulated DEGs in LT

*Figure 5 continued on next page*

*Figure 5 continued*

and LP groups. (**E**) Complement and coagulation cascades pathway between *S. parasitica* infection and linalool treatment in the spleens of grass carp. (**F**) Thirty-one DEGs related to Toll-like receptor signaling pathway. (**G**) Twenty-nine DEGs related to chemokine signaling pathway. The red squares represented upregulated genes, and the blue squares represented downregulated genes. (**H**) Expression levels of *cfh*, *masp2*, *c1r*, *c3*, *cfb*, *ap-1*, *il-1β*, and *il-6* in the spleen revealed variations among the positive control (PC), LP, and LT groups. For RT-qPCR, the results were presented as the means ± SD and were analyzed using independent *t*-tests (\*\*p < 0.01, \*\*\*p < 0.001, \*\*\*\*p < 0.0001).

epsilon RI signaling pathway, intestinal immune network for IgA production, and so on (*Figure 5D*). GO and KEGG enrichment analyses revealed the presence of numerous immune-related genes within corresponding functional processes and pathways. The expression of *fb*, *c1q*, *c2*, *c3*, *c4*, *c6*, *c7*, *c8A*, *c8b*, and *c9* genes were up expressed. Additionally, there was a significant up expressed in the gene expression levels of coagulation factors II (thrombin), VII, VIIi, VIII, X, coagulation factors IXb and IXa, serpin peptidase inhibitor clade C/D/G/F, *fga*, *fgb*, *fgg*, protein C, plasminogen, and plasminogen activator (*Figure 5E*). The differential expression of immune-related DEGs implied their role in host defense against *S. parasitica* infection.

To ensure the regulation of immune-related DEGs by linalool, the expression of several vital immune-related genes (*cfh*, *masp2*, *c1r*, *c3*, *cfb*, *ap-1*, *il-1β*, and *il-6*) in different groups were tested by RT-qPCR assay. The results showed that the expression of these genes in the LP and LT groups significantly exceeded those in the PC group (p < 0.05) (*Figure 5H*). Further exploration of immune-related genes indicated that linalool treatment promoted the expression of genes linked to the complement system and inflammatory factors. Additionally, analysis of eight genes revealed that the real-time fluorescence quantitative analysis results were consistent with the expression trend of transcriptome analysis genes.

## Linalool regulated gut microbiota composition

In our study, we explored the link between linalool and gut microbiota by conducting 16S sequencing analysis on 12 samples, which identified 862 OTUs in total, including 48 common OTUs shared across all samples (*Figure 6E*). Alpha diversity analysis (*Figure 6A–D*) showed that the Simpson, Shannon, Chao1, and Ace index of gut microbiota decreased after linalool treatment which indicated that linalool reduced the abundance and diversity of gut microbes. The goal of principal coordinate analysis (PCoA) is to preserve the distance relationships between samples as much as possible through the principal coordinates, thereby revealing the differences or patterns in microbial composition among different groups. PCoA analysis showed a significant difference in microbial composition among the PC, LP, and LT groups (*Figure 6F*).

The composition of gut microbiota in *Figure 6G* showed that the main dominant bacteria in the intestinal tract are Proteobacteria, Actinobacteriota, Verrucomicrobiota, Firmicutes, and Bacteroidota at the phylum level. Among them, compared with PC group, an obvious tendency of greater relative abundances of phyla Proteobacteria and Actinobacteriota was evidenced in LP and LT groups, the relative abundance of Verrucomicrobiota was decreased. From the composition of the genus level, the *Neochlamydia* decreased, while *Achromobacter* increased. It is worth noting that the genus *Aurantimicrobium* (classified as Actinobacteriota) was increased in LP and LT groups. Notably, among biomarkers with an LDA SCORE exceeding 3, the genus *Actinomycetales* bacterium held the highest number of positions, which contributed to the major differences between the positive control and linalool treatment groups. Moreover, the potential metabolic pathways of gut microbiota primarily showed enrichment in functions related to metabolism, including carbohydrate, amino acid, and cofactors and vitamins metabolism (*Figure 6I*).

## Correlation analysis of potential related immune genes, specific microbial species, and physical characteristics

Correlation analysis showed that complement system was significantly positively correlated to Actinobacteriota, AKP, and SR (survival rate), which, in turn, a significantly negatively correlated with Verrucomicrobiota. Additionally, there was a significant positive correlation between PPRs and AKP. Notably, IgA-related immune genes were correlated significantly with Proteobacteria (*Figure 6J*). In

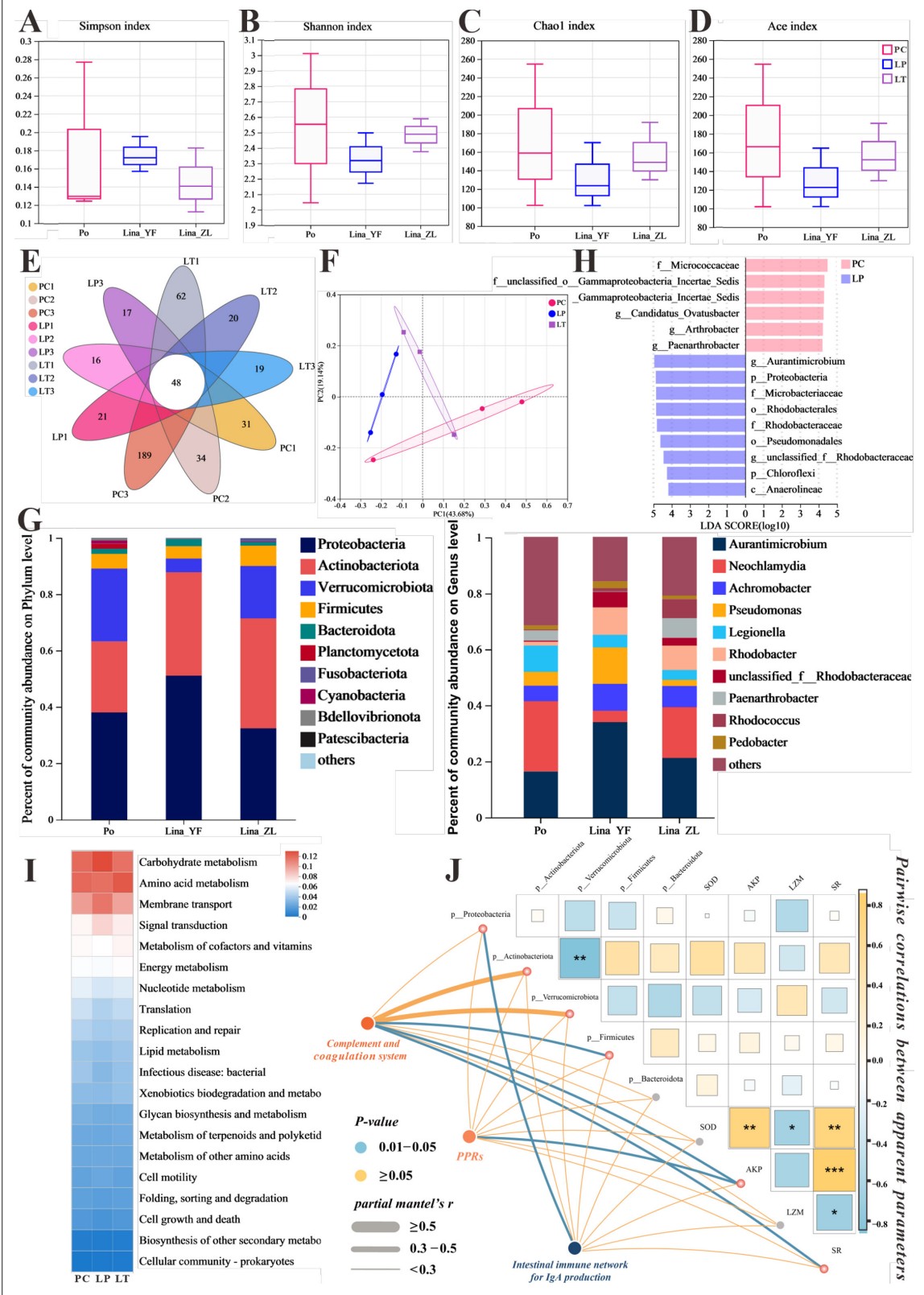

**Figure 6.** The effect of linalool on regulating gut microbiota of grass carp infected with *S. parasitica* and correlation analysis. (**A–D**) The α diversity index comparison among the different groups. (**E**) The OTUs petal map of positive control (PC), linalool prophylactic (LP), and linalool therapeutic (LT) groups. (**F**) Principal coordinate analysis (PCoA) using Bray–Curtis distance revealed variations among the PC, LP, and LT groups (ANOSIM *R* = 0.3086, p = 0.061). (**G**) Relative abundance of the top 10 species in the gut from the different groups (phylum and genus levels). (**H**) Column chart of LDA value distribution.

*Figure 6 continued on next page*

*Figure 6 continued*

Discriminative biomarkers identified by linear discriminant analysis effect size (LEfSe) with logarithmic LDA score greater than 3.0. (**I**) Heat map of the differences in predicted functional metabolisms within gut bacterial Kyoto Encyclopedia of Genes and Genomes (KEGG) pathways. (**J**) The correlation network illustrates potential associations among immune genes, microbial species, and physical characteristics. Pairwise Pearson's correlation coefficients are visualized with color gradients and block sizes, where $*|r| \geq 0.3$ with $p < 0.05$, $**|r| \geq 0.5$ with $p < 0.01$, and $***|r| \geq 0.7$ with $p < 0.001$.

summary, our findings suggested that linalool effectively modulated gut microbiota during *S. parasitica* infection which was associated with immune response.

## Discussion

### In vitro mode of action of linalool against *S. parasitica*

#### Membrane disruption

The cell membrane maintains cell integrity, regulates substance entry and exit, while the cell wall provides structural support, protection, and promotes growth and cell-to-cell connections (***Athanasopoulos et al., 2019***). The mycelium forms a dense structure to protect *S. parasitica* spores from external factors. Confocal laser scanning microscopy (CLSM), SEM, and TEM observations suggested that the cell membrane of the mycelium treated with linalool was damaged, causing the intracellular components to leak out, and could not develop completely (*Figure 2A–G*). Meanwhile, 678 genes related to the 'intrinsic component of membrane' showed significant expressed in GO enrichment analysis, suggesting that linalool may enhance the expression of intracellular membrane components, reflecting its impact on cell membrane structure or function. Gao et al. illustrated that linalool caused significant damage to the cell wall/membrane of *E. coli*, resulting in the lysis and death of bacterial cells ***Gao et al., 2023b*** demonstrated that linalool (***Gao et al., 2021***), *Changshan-huyou* Y.B. Chang EO (***Guo et al., 2019***), and Fingered Citron EO (***Zhipeng et al., 2020***) can disrupt the cell membrane of *L. monocytogenes*. Lemon EO nanoemulsions altered the permeability of *E. coli* cell membrane, affecting membrane potential, integrity, and efflux function (***Xiao et al., 2024***). Cruz et al. suggested that alkamides in *Echinacea* disrupted the cell wall/membrane complex, making it an ideal target for specifically inhibiting pathogens (***Cruz et al., 2014***). Hinokitiol caused mycelium malformation by inducing cell component leakage and influencing chitinase activity (***Meng et al., 2023***). Therefore, linalool may act anti-oomycete effects against *S. parasitica* by targeting the cell membrane.

ABC transporters utilize ATP hydrolysis energy to facilitate the export of harmful compounds or essential lipids across bacterial and eukaryotic membranes (***Ford and Beis, 2019***). The DEGs involved in ABC transporters (ABCA3, ABCB1, ABCC1, ABCG2, and PDR5) in our study, almost all of which have been linked to illness states usually involving loss of function (***Dean et al., 2001***). PDR5-related proteins in fungi played a role in excreting xenobiotics, including antifungal agents (***van den Brûle et al., 2002***). Multidrug resistance-associated protein transporters were typically associated with multidrug resistance, cellular detoxification, and drug metabolism (***Rees et al., 2009***). To sum up, linalool may greatly inhibit mycelium growth by interfering with protein synthesis from the 'inside' and damaging the integrity and permeability of the cell membrane and wall from the 'outside'.

#### Reduction of ribosome biogenesis and RNA polymerase

To further investigate the mechanism of linalool against *S. parasitica*, the gene expression profiles of *S. parasitica* were compared (with or without exposed to linalool). Many DEGs related to ribosome biogenesis and RNA polymerase were downexpressed, indicating that the linalool may influence translation, replication, and repair processes, leading to increased protein diversity and potential disruption of corresponding functions (***Chaillou et al., 2014***). Linalool influenced DNA transcription, tRNA transport, rRNA processing and maturation (5.8S, 18S, and 25S), and the biogenesis and assembly of ribosome subunits (40S and 60S) in the cell, which might lead to the reduction of *S. parasitica* growth. Ribosomes are assembled logistically by more than 200 ribosomal proteins (such as NOP, RIOK, and UTP) and RNA factors in a highly coordinated manner (***Qian et al., 2022***). Depletion of Nop-7-associated 2 (NSA2) led to decreased nascent rRNA synthesis, reduced protein synthesis, and impacted cell proliferation due to its critical role in ribosome biogenesis regulation (***Xing et al., 2018***). The inhibitory effect of myriocin on *F. oxysporum* includes both membrane damage and interactions

with intracellular targets, such as proteins like RIOK2, within the ribosome biogenesis pathway (*Wang et al., 2021c*). Innate immunity in *Caenorhabditis elegans* has been discovered to be negatively regulated by NOL-6 (also named UTP22) (*Lieschke et al., 2001*). The growth of *Aspergillus niger* may be inhibited by cyclosporin A because it may prevent the transcription and translation of the RNA polymerase III complex (*Li et al., 2023*). Rifampicin can prevent RNA transcription in bacteria by interacting with the β-subunit of DNA-dependent RNA polymerase, which can then inhibit protein biosynthesis (*Abdelwahab et al., 2016*). These results indicated that the downregulation genes may limit the proteins necessary for normal cell biogenesis and growth. Therefore, ribosome is a promising potential target of linalool against *S. parasitica*.

## The influence on the metabolism function of *S. parasitica*

Metabolism function is crucial for cells to maintain homeostasis, survive, and proliferate (*Gasmi et al., 2022*). Linalool caused the activation of energy metabolism pathways, compensating for the insufficient energy needed for cell survival (*He et al., 2023*). In our study, linalool was found to enhance amino acid metabolism, carbohydrate metabolism, energy metabolism, and so on in mycelial cells, which suggested that cells may boost functions related to energy metabolism to acquire additional energy for repairing damage caused by linalool. Hence, linalool treatment caused intracellular energy metabolism disruption, resulting in impediments to protein synthesis, cellular signal transduction, DNA synthesis, and ultimately affecting cell viability and proliferation.

## The difference in vitro mechanisms between *S. parasitica* and *S. ferax*

In our previous study, we found that linalool affected *S. ferax* mainly by altering membrane permeability, causing permanent damage, and membrane breakdown; affecting the oxidation and metabolism of fatty acids and lead to mitochondrial dysfunction and the inhibition of energy supply; degrading the branched chain amino acids and result in the dysfunction of protein synthesis. While, for *S. parasitica*, linalool exhibited antimicrobial activity by prohibiting ribosome function, inhibiting protein synthesis, disrupting the cell membrane of the mycelium. We found several interesting phenomena when compared these two species. First, the inhibition of protein synthesis was found in both species, but the interesting difference is that linalool inhibits protein synthesis through degrading the branched chain amino acids in *S. ferax*, while through prohibited ribosome function in *S. parasitica*. Second, linalool affected the oxidation and metabolism of fatty acids and mitochondrial dysfunction in *S. ferax*, while not found in in *S. parasitica*. These phenomena showing above are very interesting results and we are eager to explore the biological mechanisms in the future.

## In vivo mode of action of linalool against *S. parasitica* infection

Recent studies demonstrated that the spleen serves antibacterial functions in various fish species, with a high quantity of melano-macrophages that assist in phagocytosis during the immune response (*Gao et al., 2023a*). The rise of macrophages and the change in the pathological and physiological circumstances in fish are correlated (*Xu et al., 2018*). Similarly, histopathological analysis demonstrated that linalool effectively reduced *S. parasitica* infection-induced damage in the spleen, indicating its potential immunomodulatory effects on immune organs. Here, we used comparative transcriptome analysis to investigate potential genes and signaling pathways regulated by linalool in the spleen of grass carp. Several genes associated with both innate and adaptive immunity, such as complement molecules, pattern recognition receptors, inflammatory cytokines, immunoglobulins, and chemokines, were differentially expressed as shown in *Figure 5E–G*.

## Enhancement of complement and coagulation system

It is widely acknowledged that the complement and coagulation system in fish is an important immune defense mechanism for fighting pathogens and maintaining immune homeostasis (*Oikonomopoulou et al., 2012*). The complement system is made up of over 30 different proteins (complement receptors, internal components, and regulatory proteins) and 3 activation pathways (the classic pathway, the lectin pathway, and the alternative pathway) (*Pan et al., 2022*). In our study, the expression of *fb*, *c1q*, *c2*, *c3*, *c4*, *c6*, *c7*, *c8a*, *c8b*, and *c9* genes were all upregulated after LT. Meanwhile, the *c3a* anaphylatoxin chemotactic receptor (*c3aar1*) and mannan-binding lectin serine protease 2 (*masp2*) genes were up expressed. During complement activation, *c3* and *c5* split into molecules such as *c3a*, *c3b*, *c5a*,

and *c5b*, of which *c3a* and *c5a* are considered strong inflammatory mediators because of their ability to attract leukocytes and promote inflammation and pathogen clearance (*Kobayashi et al., 2016*). *masp2* activates complement components such as *c4* and *c2*, triggering a complement cascade reaction that ultimately leads to pathogen rupture (*Krarup et al., 2007*). *c8b* is a subunit of the complement component 8 (*c8*) protein, which helps form the membrane attack complex for cell lysis and immune defense against microbes (*Xie et al., 2022*). In addition, high expression of complement factor H like 3 (*chfl3*) prevents overactivation of the complement system to avoid damage to their own tissues (*Ferluga et al., 2017*). The effective antimicrobial role of the *c5a–c5ar1* signaling pathway in the complement system has been identified. It functions by enhancing neutrophil phagocytosis of fungi and macrophage-mediated fungicidal activity to combat fungal infections (*Desai et al., 2023*). In all, our findings suggested that linalool may enhance the complement system which in turn activated host immune defense and lysate *S. parasitica* cells.

The coagulation cascades are connected with the complement system. Linalool significantly increased the expression of fibrinogen alpha (*fga*), beta (*fgb*), and gamma (*fgg*) chain, which together encoded fibrinogen (*Uitte de Willige et al., 2005*), fibrinogen is a crucial component of blood clots to prevent blood loss, and also important in the initial stages of wound healing (*Wang et al., 2021a*). Second, activated prothrombin *f2* is central in clot formation, platelet aggregation, and activating other clotting factors. It also supports tissue repair, cell growth, angiogenesis, and vascular integrity (*Xie et al., 2022*). Third, an increase in plasminogen levels can cause an immune response (*Bryer et al., 2008*), and serve as a regulator of the natural immune system by promoting the phagocytosis of phagocytes (*Das et al., 2014*). Taken together, the upregulation of fibrinogen, prothrombin, and plasminogen in the coagulation system suggested that linalool may promote wound healing, tissue repair, and phagocytosis to cope with *S. parasitica* infection.

## Regulation of gut microbiota and increase the abundance of Actinobacteriota

Gut microbiota controls host growth and health status through various pathways (*Zhu et al., 2022*). First, in our study, the function of gut microbiota is mainly enriched in metabolic-related pathways linalool increased the abundance of the beneficial phylum Actinobacteriota and decreased the abundance of the harmful phylum Verrucomicrobiota. Actinobacteriota plays a crucial role because Actinobacteria can produce enzymes, antibiotics, short-chain fatty acids, signaling molecules, and immunomodulators (*Xin et al., 2022*). Meanwhile, Actinobacteriota correlated positive with the complement and coagulation system according to the correlation analysis. Second, intestinal IgA production was correlated significantly with Proteobacteria which are the primary inducers of IgA production by B cells (*Mirpuri et al., 2014*). In summary, our findings suggested that linalool may positively modulate the immune response by increasing the abundance of beneficial phylum Actinobacteriota to *S. parasitica* infection.

## Regulation of inflammatory factors

Pattern recognition receptors detect and initiate multiple signaling pathways upon recognizing pathogen-associated molecular patterns on the surface of pathogens (bacteria, viruses, fungi, etc.), inducing subsequent host immune responses (*Khan et al., 2019*). Myeloid differentiation primary-response protein 88 (MyD88) serves as the crucial signaling adaptor protein for Toll-like and interleukin-1 receptors, which is responsible for the conduction of multiple downstream signaling pathways (*Minderman et al., 2023*). The expression of *ap-1*, *p-38*, and *pI3k* were upregulate in downstream signaling pathways, causing higher expression of inflammatory factors, and enhanced cellular oxidation and antioxidant levels (*Wang et al., 2023*). Meanwhile, in our study, the gene expression levels of *il-1β*, *il-6*, *ccl19*, and *ccl5* which are closely related to the inflammasome, were up expression. Inflammasomes are protein complexes capable of detecting diverse inflammatory stimuli and are vital for eliminating pathogens or injured cells (*Li et al., 2021*). Therefore, linalool might have influenced the immune system response in grass carp, regulating the inflammatory processes.

## Conclusions

In summary, our study proved that linalool is an effective natural antimicrobial agent against *S. parasitica*. Linalool disrupted the cell membrane, resulting in the leakage of cellular components and

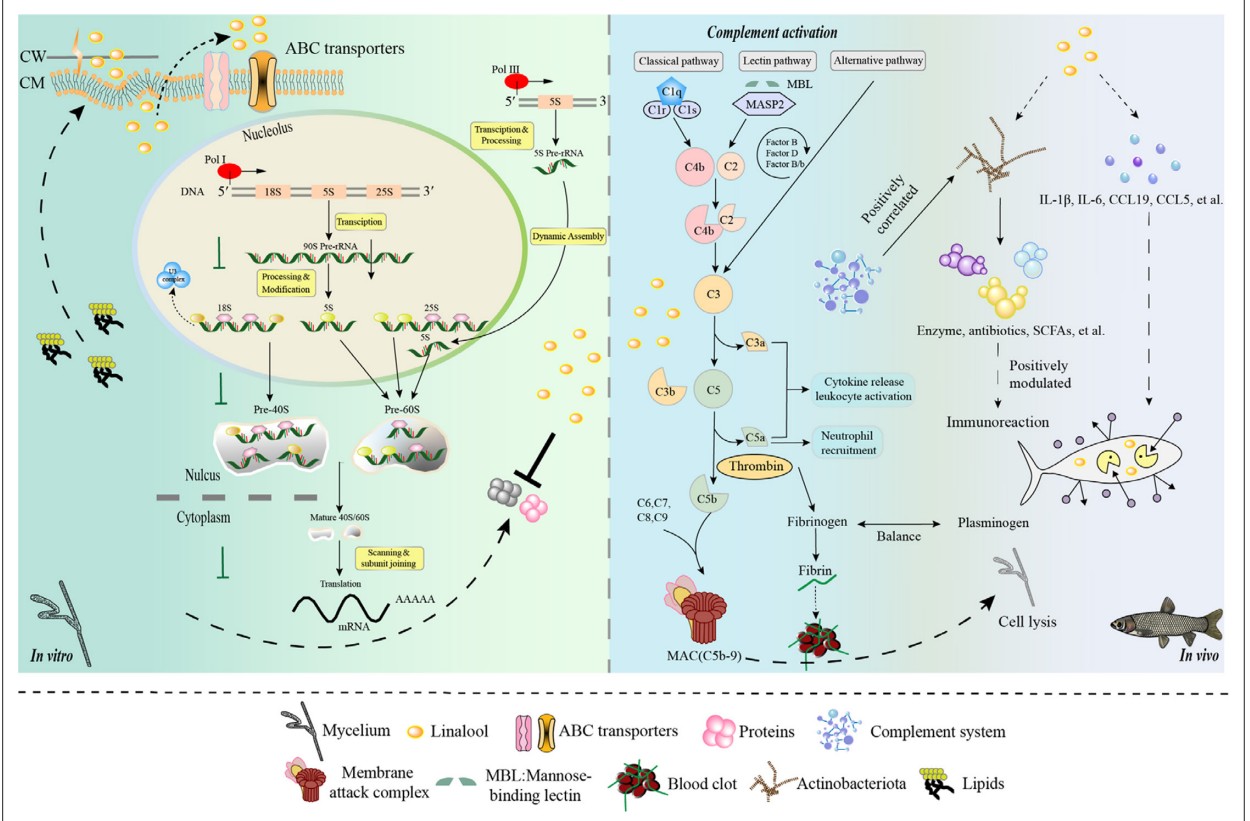

**Figure 7.** Model diagram of the mode of action of linalool on *S. parasitica* and grass carp. In vitro, (1) Linalool influenced DNA transcription, tRNA transport, rRNA processing and maturation (5.8S, 18S, and 25S), and the biogenesis and assembly of ribosome subunits (40S and 60S) in the cell, which might lead to the reduction of S. parasitica growth; (2) Linalool disrupted the cell membrane, and the upregulation of glycerophospholipid metabolism likely represents the cell's response to cope with this damage; (3) ABC transporters contributed to metabolic resistance by pumping linalool out of the cell. In vivo, (1) Linalool enhanced the complement and coagulation system which in turn activated host immune defense and lysate *S. parasitica* cells; (2) Linalool promoted wound healing, tissue repair, and phagocytosis to cope with *S. parasitica* infection; (3) Linalool positively modulated the immune response by increasing the abundance of beneficial Actinobacteriota; (4) Linalool stimulated the production of inflammatory cytokines (*il-1β* and *il-6*) and chemokines (*ccl19* and *ccl5*) to lyse *S. parasitica* cells.

restricting ribosomal function, thereby inhibiting protein synthesis. Comprehensive analyses revealed the specific role of linalool in protecting *S. parasitica* infection skin inflammation in grass carp. Meanwhile, the complement and coagulation systems, inflammatory factors, along with Actinobacteriota, significantly involved in this protective effect (*Figure 7*). Further investigation would be carried out to explore and refine how linalool affects the inflammatory effects of *S. parasitica* on fish through signaling pathways and mechanisms. We believe that this study will contribute to developing an innovative alternative strategy, distinct from antibiotics, to cope with *S. parasitica* infection in the aquaculture industry.

## Materials and methods
### Oomycete strains and materials
*S. parasitica* used in this study was from our lab, which was isolated from grass carp (*Ctenopharyngodon idella*). Linalool (0.862 g/ml), pinene (0.874 g/ml), limonene (0.842 g/ml), myrcene (≥95%), terpinene (≥95%), resazurin dye, and PI were purchased from Sigma-Aldrich (USA). SYTO 9 and FM4-64 were purchased from Thermo Fisher (USA). PDA and glucose yeast extract (GY) were purchased from Guangdong Huankai Microbial Sci. and Tech. Co., Ltd, China.

## Agar diffusion assay

The anti-oomycetes effects of EOs and five components (linalool, terpinene, limonene, myrcene, and pinene) were assessed by using the agar diffusion method. Initially, a 6-mm mycelium plug was inoculated at the center of the plates. Subsequently, 6-mm sterile filter paper discs were soaked in EOs and components for more than 12 hr and placed symmetrically at a distance of 2.5 cm from the target mycelium plug. Sterile distilled water was used as NC. Finally, the width of the colonies was measured once the mycelium in the control group was completely covered the plate.

Inhibition rate (%) = [(P1 − P2)/P1] × 100%, where P1 = colony growth diameter (mm) of control group and P2 = colony growth diameter (mm) of treated group.

## Minimum inhibitory concentration and MFC assays

To determine the minimum inhibitory concentration (MIC) and MFC of linalool on mycelium, 6 mm mycelium plugs were inoculated into a 24-well cell culture plate. The wells contained linalool at concentrations of 0.8%, 0.4%, 0.2%, 0.1%, 0.05%, 0.025%, 0.0125%, and 0.00625%, dissolved in 1% Tween 20 and potato dextrose broth. Wells without mycelium plugs were served as negative controls. After incubating at 25°C for 48 hr, the lowest concentration of linalool with no visible mycelium growth was recorded as MIC. Mycelium plugs showing no growth at this MIC concentration then inoculated onto PDA plates and incubated for 7 days at 25°C to determine MFC.

The method for spores induction is similar as previously described by our previous research (**Tang et al., 2024**). In a 96-well cell culture plate, 100 μl of linalool at concentrations of 0.4%, 0.2%, 0.1%, 0.05%, 0.025%, 0.0125%, 0.00625%, and 0.003125% were dissolved in 1% Tween 20 and GY were added initially to each well, followed by the addition of 100 μl of spore suspension at a concentration of $1 \times 10^6$ spores/ml to each well. 2.5 mg/ml malachite green and 1% Tween 20 served as positive and negative controls, respectively. Following a 24-hr incubation at 20°C, 5 μl of resazurin dye was added. When the color of the wells turned to pink, the linalool concentration in the first non-pink well corresponded to the MIC. The culture liquid from wells with no visible mycelium growth was streaked onto new plates. After incubating at 25°C for 7 days, the well with the lowest linalool concentration with no visible mycelium growth was determined as MFC.

## Effect of linalool on the mycelium radial growth rate of *S. parasitica*

Different concentrations of linalool were added to molten PDA plates at 50°C for quantitative assessment. A similarly sized mycelium plug was placed at the center of each plate. Once the mycelium in the control group without linalool reached the edge of the plate (approximately 60 hr), the radial diameter of the mycelium was measured.

The calculation of growth inhibition was determined by assessing the percentage inhibition of radial growth, as below:

The percentage inhibition of radial growth (PIRG %) = [(R1 − R2)/R1] × 100%, where R1 = radial growth of control plate (mm) and R2 = radial growth of treatment plate (mm).

## CLSM assay

### Analysis of viability of *S. parasitica*

The mycelium was treated with linalool at MIC (0.05%) and 2× MIC (0.1%) for 2 hr, and untreated control group was prepared simultaneously. The mycelium resuspended in PBS was further treated with PI at a final concentration of 2.5 g/ml for 15 min. PI is excited at a wavelength of 488 nm and emits light at 620 nm. Fluorescence images were observed using CLSM (LSM880, Zeiss, Germany) at ×40 magnification.

### Analysis of membrane damage of *S. parasitica*

Mycelium cultured for 18 hr was collected and sequentially treated with SYTO 9 at a concentration of 20 μmol/ml, FM4-64 at a concentration of 10 μmol/ml, and linalool at a concentration of 3%. Record observations at 0, 2, 8, and 20 min. Both SYTO 9 and FM4-64 have an excitation wavelength of 488 nm and emit light at 620 and 520 nm, respectively. Fluorescence images were observed using CLSM (LSM880, Zeiss, Germany) at ×40 magnification.

## SEM and TEM assays

Mycelium exposed to MIC (0.05%) linalool for 2 hr underwent fixation with 2.5% glutaraldehyde at 4°C for 4 hr to examine the impact of linalool on membrane damage. Following a rinse with phosphate-buffered saline (PBS; pH 7.2), the fixed samples experienced dehydration for 15 min using a graded series of ethanol concentrations (10%, 30%, 50%, 70%, 90%, and 100%), followed by an additional 20 min in 100% ethanol. Post-dehydration, ethanol was exchanged twice for 20 min each with tertiary butanol. Subsequently, the samples were subjected to vacuum freeze-drying, gold-spraying, and examination with SEM (Hitachi, SU8010). After fixation, the samples were ultrathin-sectioned, subjected to negative staining using 1% phosphotungstic acid for 5 min and observed by TEM (Hitachi, HT-7700).

## RNA-Seq and bioinformatics analysis

Total RNA was isolated from linalool treated (exposed to 0.5× MIC linalool for 10 min) and untreated control groups using the TRIzol method. The messenger RNA (mRNA) was enriched and fragmented into 200 bp fragments. These mRNA fragments were reverse-transcribed into cDNA using random primers and reverse transcriptase. The cDNA ends were modified and ligated to an adaptor. After PCR amplification and purification, Illumina sequencing was performed. Quality control removed low-quality reads. The mapped reads were assembled using StringTie. Gene expression was quantified with the transcripts-per-million-reads method to identify DEGs using a threshold of |log twofold change| ≥2 and FDR <0.05. Afterward, functional enrichment analysis was performed using the KEGG database. KEGG pathway analysis was conducted using Goatools. The Majorbio cloud platform was used for data analysis (*Guo et al., 2021*).

## Molecular docking

Linalool may interact with specific domains of the ribosome, potentially influencing its structure and function. Therefore, our investigation aimed to determine whether linalool could bind to the ribosome-associated upstream proteins of *S. parasitica*. As no crystal structure of *S. parasitica* ribosome proteins existed in the Protein Data Bank (PDB), we performed homology modeling. The protein sequences NOP1 (XP_012204452.1), SNU13 (XP_012198008.1), and DKC1 (XP_012193765.1) were obtained from GenBank, and homology modeling was conducted using AlphaFold2. The most optimal configuration identified from the homologous modeling results was chosen as the receptor protein for subsequent molecular docking. The model was then evaluated using ProSA-web and PROCHECK server. The secondary structure of linalool was then loaded on PubChem. Both the ligand (linalool) and target proteins underwent processing with AutoDockTools 1.5.6, including energy minimization, dehydration, and hydrogenation. The conformation exhibiting the lowest energy was selected as the most likely binding state. The ultimate docking outcome was visualized utilizing the open-source software PyMOLv2 and Discovery Studio Visualizer V2021.

## Animal experiments

### Linalool therapeutic and prophylactic assays

In this study, four groups were established: (1) PC group (10 fish infected with *S. parasitica*). (2) LT group (10 fish infected with *S. parasitica*, soaked in 0.00039% linalool in a 20-l tank for 7 days). (3) LP group (10 uninfected fish soaked in 0.00039% linalool in a 20-l tank for 2 days, followed by the addition of 1 × 10⁶ spores/ml secondary zoospores). (4) NC group (10 uninfected fish without linalool treatment). Each group had three replicate tanks. The number of fish without visible mycelium on day 7 and the cumulative survival rate were recorded. The fish samples from the NC, PC, LT, and LP groups were collected for histopathological examinations. Tissue samples underwent fixation, sectioning, hematoxylin and eosin staining, and microscopic examination using a microscope (BX53, Olympus).

### 16S rRNA and transcriptome sequencing and analysis

Genomic DNA was extracted from grass carp intestinal content samples and stored at –20°C following the instructions of the 16S rRNA gene pyrosequencing DNA kit. We conducted paired-end sequencing on the Illumina MiSeq platform, amplifying the V3–V4 regions with specific barcoded primers using a thermocycler PCR system. To unveil the intricate microbial diversity within the samples, we employed Quantitative Insights Into Microbial Ecology (QIIME) which can compute and visualize α-diversity, β-diversity, and PCoA. α-Diversity measured the species richness and evenness within a single sample,

reflecting the complexity of the community; β-diversity compared the differences in species composition between samples, revealing changes in community structure; PCoA, based on a distance matrix, visualized the differences between samples, providing an intuitive display of community clustering and distribution. Each sample was measured three times to ensure robust results.

Spleen tissue was obtained from three fish in each group on day 7. These collected spleen tissues were promptly flash-frozen in liquid nitrogen and then stored at –80°C until they were ready for RNA isolation. RNA-Seq and bioinformatics analysis was carried out as shown in the section 'Linalool regulated gut microbiota composition'.

In each group, eight fish were utilized for immunological assays, and on day 7, blood samples were collected from the tail veins using heparinized syringes and left to coagulate overnight at 4°C. After centrifugation at 3500 × *g* for 10 min at 4°C, serum was obtained from each fish and stored at –80°C for subsequent analysis. Kits from Nanjing Jiancheng Institute (Nanjing, China) were used to measure lysozyme activity, SOD activity, and AKP activity.

## Detection of immune-related gene expression

Spleen tissues RNA extraction utilized the EZNA Total RNA Kit II (OMEGA R6934-02). RNA concentrations were quantified using a spectrophotometer (Eppendorf BioSpectrometer basic, Germany), and quality assessment was conducted through 1% gel electrophoresis. Subsequent cDNA synthesis employed reverse transcription with the RevertAid First Strand cDNA Synthesis Kit (Thermo Fisher Scientific, USA). RNA concentrations were quantified via spectrophotometry (Eppendorf BioSpectrometer basic, Germany). Subsequently, cDNA was synthesized via reverse transcription using the RevertAid First Strand cDNA Synthesis Kit (Thermo Fisher Scientific, USA). Expression levels of complement factor H (*cfh*), mannan-binding lectin serine protease 2 (*masp2*), complement component 1, r subcomponent (*c1r*), complement component 3 (*c3*), complement factor B (*cfb*), transcription factor AP-1 (*ap-1*), interleukin 1 beta (*il-1β*), and interleukin 6 (*il-6*) were detected. The PCR procedure was described as follows: 95°C for 30 s, followed by 40 cycles of 95°C for 5 s and 60°C for 40 s. Employing the CFX96 Touch Real-time PCR Detection System (Bio-Rad, USA), gene expression levels

**Table 1.** Primers for qPCR were used in this study.

| Primer name | Primer sequence 5'–3' |
| --- | --- |
| *c3*-F | ACAAGCCCATCTACACGCCT |
| *c3*-R | GAAACCTCAGGAATGGCGTA |
| *cfb*-F | AGAATGGAGAGGTGACCCCA |
| *cfb*-R | CCGCTATCCTGACACACTCG |
| *c1r*-F | GGGTTTCGGAGGGTTTGA |
| *c1r*-R | CCTGTATTGCCACATCTTTCA |
| *cfh*-F | ATAGGAGGGAAATGTGGGC |
| *cfh*-R | TCCCGTCCACGCCATC |
| *masp2*-F | TGGGCACCAGGTTCATCG |
| *masp2*-R | GCTGTTTGACTACGCCCATCTT |
| *il-1b*-F | ATCTCCTCGTCTGCTGGGTG |
| *il-1b*-R | TTGAGAGTTTCAGTGACCTCCTTC |
| *il-6*-F | AAAATCTGACCTCAACCCTGG |
| *il-6*-R | TGTCCACCCTTCCTCTTGCT |
| *ap-1*-F | TCTCACTCAACAGATGCCAAAC |
| *ap-1*-R | CTTAGCGGCTGCGAGTTTAT |
| β-ActinF | GCTATGTGGCTCTTGACTTCG |
| β-ActinR | GGGCACCTGAACCTCTCATT |

were assessed using the $2^{-\Delta\Delta Ct}$ method, with β-actin serving as the reference gene. The primers in qPCR analysis are listed in *Table 1*.

## Statistical analysis

The data were expressed as mean ± standard error, and all experiments were conducted in triplicate. Pearson correlation analysis was employed to investigate the correlation between potential immune-related genes in grass carp and their characteristics, the method was implemented in R Programming language. Statistical analysis was performed using GraphPad Prism 9 software, and the Student *t*-test was conducted.

## Acknowledgements

This research was funded by Hunan Provincial Natural Science Foundation (2025JJ50171, 2023JJ40364), National Natural Science Foundation of China (32073020, 32201960), and Science and Technology Innovation Program of Hunan Province (2022RC1150).

## Additional information

### Funding

| Funder | Grant reference number | Author |
|---|---|---|
| Hunan Provincial Natural Science Foundation of China | 2025JJ50171 | Jiajing Guo |
| Hunan Provincial Natural Science Foundation of China | 2023JJ40364 | Jiajing Guo |
| National Natural Science Foundation of China | 32073020 | Zhipeng Gao |
| National Natural Science Foundation of China | 32201960 | Jiajing Guo |
| Science and Technology Innovation Program of Hunan Province | 2022RC1150 | Jiajing Guo |
| Agricultural Science and Technology Innovation Fund of Hunan | 2023CX49 | Jiajing Guo |

The funders had no role in study design, data collection, and interpretation, or the decision to submit the work for publication.

### Author contributions

Tao Tang, Data curation, Software, Formal analysis, Validation, Investigation, Visualization, Methodology, Writing – original draft; Weiming Zhong, Software, Investigation; Puyu Tang, Investigation, Methodology; Rongsi Dai, Visualization, Methodology; Jiajing Guo, Conceptualization, Supervision, Writing – review and editing; Zhipeng Gao, Conceptualization, Supervision, Funding acquisition, Project administration, Writing – review and editing

### Author ORCIDs

Tao Tang ⓘ https://orcid.org/0009-0008-1320-0227
Zhipeng Gao ⓘ https://orcid.org/0000-0001-7787-285X

### Ethics

All experiments involving animals were conducted according to the ethical policies and procedures approved by the Institutional Animal Care and Use Committee of Hunan Agricultural University, China (Approval No. 430516).

Reviewer #1 (Public review): https://doi.org/10.7554/eLife.100393.3.sa1
Reviewer #2 (Public review): https://doi.org/10.7554/eLife.100393.3.sa2
Author response https://doi.org/10.7554/eLife.100393.3.sa3

## Additional files

### Supplementary files
MDAR checklist

### Data availability
The 16S rRNA and RNA-Seq data were uploaded to the National Center for Biotechnology Information SRA database as part of BioProject PRJNA1235868 and PRJNA1235867, respectively.

The following datasets were generated:

| Author(s) | Year | Dataset title | Dataset URL | Database and Identifier |
|---|---|---|---|---|
| Tang T, Zhong W, Tang P, Dai R, Guo J, Gao Z | 2025 | SRA database | https://www.ncbi.nlm.nih.gov/bioproject/PRJNA1235868 | NCBI BioProject, PRJNA1235868 |
| Tang T, Zhong W, Tang P, Dai R, Guo J, Gao Z | 2025 | SRA database | https://www.ncbi.nlm.nih.gov/bioproject/PRJNA1235867 | NCBI BioProject, PRJNA1235867 |

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
