## [Editor Report · eLife Assessment]

This **useful** study presents findings on the efficacy and mechanisms of linalool protection against Saprolegnia parasitica oomycetes in the grass carp model. The evidence presented is **solid** since the methods, data and analyses broadly support the claims with only minor weaknesses. This work will be of great interest to scientists within the fields of aquaculture, ichthyology, microbiology, and drug discovery.

---

## [Referee Report · Reviewer #1 (Public review)]

Summary:

The works seeks to investigate the efficacy of linalool as a natural alternative for combating Saprolegnia parasitica infections, which would provide great benefit to aquaculture. This paper shows the effect of linalool in vitro using a variety of techniques including changes in S. parasitica membrane integrity following linalool exposure and alterations in cell metabolism and ribosome function. Additionally, this work goes on to show that prophylactic and concurrent treatment of linalool at the time of S. parasitica infection can improve survival and tissue damage in vivo in their grass carp infection model. The conclusions of the paper are partially supported by the data with the corrections done by the authors improving clarity such that I believe there is merit in the work.

---

## [Referee Report · Reviewer #2 (Public review)]

Summary:

In this study, the authors aimed to delineate the antimicrobial activity of linalool and tried to investigate the mode of action on linalool against S. parasitica infection. One of the main focus of this work was to identify the in vitro and in vivo mechanisms associated with the protective role of linalool against S. parasitica infection.

Strengths:

(1) Authors have used a variety of techniques to prove their hypothesis.

(2) Adequate number of replicates were used in their studies.

(3) Their findings showed a protective role of linalool against oomycetes and makes it an attractive future antibiotic in the aquaculture industry.

Weaknesses: The revised version of the manuscript is more thoroughly written with clearer explanations, however there are a few weaknesses in this manuscript.

(1) Although the introduction section was rewritten with rationale, it's still lengthy and not very much to the point.

(2) The claim of linalool regulating the gut microbiota is based on the correlation analysis only. It's not super convincing and requires experimental validation to strengthen the claim.

Overall, the conclusions drawn by the authors are justified by the data. Importantly, this paper has discovered the novelty of the compound linalool as a potent antimicrobial agent and might open up future possibilities to use this compound in the aquaculture industry.

---

## [Author Response]

The following is the authors’ response to the original reviews.

**Reviewer #1 (Public review):**
(1) Adding microscopy of the untreated group to compare Figure 2A with would further strengthen the findings here.

First of all, we would like to thank Reviewer #1 for their comments and efforts on our manuscript. We have carefully revised it. We used a time-lapse method to capture images at 0 minutes, before any drugs were added. We will change '0 min' to 'untreated,' which will further strengthen the findings.

(2) Quantification of immune infiltration and histological scoring of kidney, liver, and spleen in the various treatment groups would increase the impact of Figure 4.

Thank you very much to Reviewer #1 for their comments and efforts on our manuscript. We have revised it carefully. We conducted quantitative analysis of immune infiltration in the kidney, liver, and spleen across different treatment groups. However, due to the extremely low number of abnormal cells in the negative control, treatment, and prophylactic groups, neither the instrument nor manual methods could reliably gate the cells. Consequently, quantification of immune infiltration and histological scoring were not performed.

(3) The data in Figure 6 I is not sufficiently convincing as being significant.

Thanks so much for Reviewer #1 comments and efforts for our manuscript. We have revised it carefully. Previous researches have shown that antibiotics and other drugs can cause alterations in gut microbiota. Therefore, we plan to study the effects of antibiotics on gut microbiota. To conduct this research, we need to isolate these microbes from the gut. Although this process is challenging, we still aim to explore the gut microbiota. If possible, we will continue to delve into interesting aspects of how antibiotics affect gut microbiota in future studies.

(4) Comparisons of the global transcriptomic analysis of the untreated group to the PC, LP, and LT groups would strengthen the author's claims about the immunological and transcriptomic changes caused by linalool and provide a true baseline.

Thanks so much for Reviewer #1 comments and efforts for our manuscript. We have revised it carefully. Due to the initial research design and data analysis strategy, we have focused on comparisons among the PC, LP, and LT groups to more directly explore the differences under various treatment conditions. Specifically, while the transcriptomic data from the untreated group could provide a basic reference, it has shown limited relevance to the core hypotheses of our study. Our research has aimed to investigate the immunological and transcriptomic changes among the treatment groups rather than comparing treated and untreated states. We believe that the current experimental design and data analysis have effectively revealed the mechanisms of linalool and that the additional comparisons among the treatment groups have further supported our conclusions. We hope the reviewer understands the rationale behind our experimental design. If there are additional suggestions, we are more than willing to further optimize the content of our manuscript.

**Reviewer #2 (Public review):**
(1) The authors have taken for granted that the readers already know the experiments/assays used in the manuscript. There was not enough explanation for the figures as well as figure legends.

Thanks so much for Reviewer #2 comments and efforts for our manuscript. We have revised it carefully. We will provide more detailed explanations of the experiments and assays used in the manuscript, as well as enhance the descriptions in the figure legends, to ensure that readers have a clear understanding of the figures and their context.

(2) The authors missed adding the serial numbers to the references.

Thanks so much for Reviewer #2 comments and efforts for our manuscript. We have revised it carefully. We will add serial numbers to the references to ensure proper citation and improve the clarity of our manuscript.

(3) The introduction section does not provide adequate rationale for their work, rather it is focused more on the assays done.

Thanks so much for Reviewer #2 comments and efforts for our manuscript. We have revised it carefully. We will add a section to the introduction that provides a rationale for our work, specifically focusing on the impact of plant extract on immunoregulation.

(4) Full forms are missing in many places (both in the text and figure legends), also the resolution of the figures is not good. In some figures, the font size is too small.

Thanks so much for Reviewer #2 comments and efforts for our manuscript. We will ensure that all abbreviations are expanded where necessary, both in the text and figure legends. Additionally, we will improve the resolution of the figures and increase the font size where needed to enhance clarity.

(5) There is much mislabeling of the figure panels in the main text. A detailed explanation of why and how they did the experiments and how the results were interpreted is missing.

Thanks so much for Reviewer #2 comments and efforts for our manuscript. We have revised it carefully. We will improve the labeling of the figure panels, provide detailed explanations of the experimental methods, including their rationale and interpretation, and clarify the connections between the methods.

(6) There is not enough experimental data to support their hypothesis on the mechanism of action of linalool. Most of the data comes from pathway analysis, and experimental validation is missing.

Thanks so much for Reviewer #2 comments and efforts for our manuscript. We have revised it carefully. Actually, in our manuscript the transcriptomic data are not alone, and we carried out many experiments to substantiate the changes inferred from the transcriptomic data as SEM, TEM, CLSM, molecular docking, RT-qPCR, histopathological examinations. The detailed information is listed as below.

As shown in Figure 2, we combined the transcriptomic data related to membrane and organelle with SEM, TEM, and CLSM images. After deep analysis of these data and observation together, we illustrated that cell membrane may be a potential target for linalool.

As shown in Figure 3, we carried out molecular docking to explore the specific binding protein of linalool with ribosome which were screen out as potential target of linalool by transcriptomic data.

As shown in Figure 5, transcriptomic data illustrated that linalool enhanced the host complement and coagulation system. To substantiate these changes, we carried out RT-qPCR to detect those important immune-related gene expressions, and found that RT-qPCR analysis results were consistent with the expression trend of transcriptome analysis genes.

As shown in Figure 4 and 5, transcriptomics data revealed that linalool promoted wound healing tissue repair, and phagocytosis (Figure. 5E). To ensure these, we carried out histopathological examinations, and found that linalool alleviated tissue damage caused by *S. parasitica* infection on the dorsal surface of grass carp and enhancing the healing capacity (Figure. 4G).

Overall, we will conduct additional experiments to verify the mechanism of action of linalool in the future.

**Reviewer #1 (Recommendations for the authors):**
(1) Figure 1 Panel G is not referenced in the legend, this should be fixed

Thanks so much for Reviewer #1 comments and efforts for our manuscript. We have revised it carefully. Please check the Figure 1. The order of Panel F and G in Figure 1 is wrong. We have modified the order of Figure 1.

(2) Statistical comparisons between groups in Figure 4 Panels C-F is lacking and should be added.

Thanks so much for Reviewer #1 comments and efforts for our manuscript. We have revised it carefully. Please check the Figure 4 C-F. We have added statistical comparisons between groups in Figure 4 Panels C-F.

(3) Capitalize Kidney label in Figure 4G.

Thanks so much for Reviewer #2 comments and efforts for our manuscript. We have revised it carefully. Please check the Figure 4G. We have capitalized the K of kidney.

**Reviewer #2 (Recommendations for the authors)**:(1) The authors missed adding the serial numbers to the references. I could not go through the references to cross-check if they cited the right ones because it's extremely difficult to figure out which one corresponds to which reference number.

Thanks so much for Reviewer #2 comments and efforts for our manuscript. We have revised it carefully. Please check the references. We have added the serial numbers to the references.

(2) In the last paragraph of the introduction section, most of the techniques in the paper were summarized which does not go with the flow of the paper. The introduction should not be focused on the different techniques used the focus should be more on the rationale of the work. It would be nice if the last paragraph could be rewritten.

Thanks so much for Reviewer #2 comments and efforts for our manuscript. We have revised it carefully. Please check it in Line 85-94. We have added a section to the introduction that provides a rationale for our work, specifically focusing on the impact of plant extract on immunoregulation.

(3) The resolution of the figures is not good.

Thank you for your suggestion. We have revised it carefully. Please check all the figures. We have increased the resolution and size of all the figures.

(4) Mostly, the figure legends sound like results, with not enough explanation. Full forms are missing in many places which would make the readers go back to the text/other figures each time.

Thanks so much for Reviewer #2 comments and efforts for our manuscript. We have revised it carefully. Please check it throughout the manuscript and all the figure legends. We have added full names and abbreviations to both the manuscript and all the figure legends so that we don't make the readers go back to the text/other figures each time.

(5) Figure 1:Figure 1A: there is not enough explanation for this panel. It's not clear from the text which other EOs than Linalool are referred to here. Which EOs were extracted from daidai flowers?

Thanks so much for Reviewer #2 comments and efforts for our manuscript. We have revised it carefully. Please check it in the Figure 1A. Figure 1A is divided into “Essential oils (EOs)” and “The main compounds of EOs” to make it easier to distinguish.

Figure 1B: do the three different wells of each set represent three replicates? If so, are they biological/technical replicates? Also, I'm not sure how the MFC was determined from this figure (line 116) because clearly this panel only corresponds to the determination of MICs, not MFCs.

Thanks so much for Reviewer #2 comments and efforts for our manuscript. We have revised it carefully. Please check it in Line 126-130. The three different wells of each set represent three biological replicates. After adding 5 μL of resazurin dye, when the color of the wells turned to pink, the linalool concentration in the first non-pink well corresponded to the MIC. The culture liquid in the well where no mycelium growth was seen was marked onto the plate and incubated at 25°C for 7 days. The well with the lowest linalool concentration and no mycelium growth was identified as MFC.

Figure 1C: the figure legend says that the effect of linalool on mycelium growth inhibition was done over a 6hr timepoint but according to the figure the timepoint was 60hr. I am also confused about the concentrations of linalool used. Although a range of concentration from 0 to 0.4% is mentioned, I only see the time vs diameter curves for 7 concentrations.

Thanks so much for Reviewer #2 comments and efforts for our manuscript. We have revised it carefully. Please check it in Line 983 and Figure 1C. We have changed 6 h to 60 h in the figure legends. The reason why only the time vs diameter curves for 7 concentrations in Figure1C is that the growth inhibition of 0.4%, 0.2% and 0.1% linalool on mycelial growth is the same. As a result, the time vs diameter curves coincide. We have shown the time and diameter curves of 0.4%, 0.2% and 0.1% concentration with three dotted lines of different colors and sizes in Figure 1C.

Figure 1D: mislabeled as 1G in the figure panel.Figures 1E and 1G: Figure 1E is missing and I do not see any figure legend for Figure 1G.

Thanks so much for Reviewer #2 comments and efforts for our manuscript. We have revised it carefully. Please check the Figure 1. The order of Panel F and G in Figure 1 is wrong. We changed the order of Figure 1 ABCDEF, no Figure G.

Overall, Figure 1 is very confusing and needs rewriting. Also, there is a need to add more explanation of the figure panels in the results section.

Thanks so much for Reviewer #2 comments and efforts for our manuscript. We have revised it carefully. Please check the Figure 1. We have corrected all the problems in Figure1. And we have added more explanation of the figure panels in the results section, and increased the correlation between methods, in order to show how to carry out the experiment logically and interpret the results, please check them in Line 126-130, 144-147, 174-179, 213-217, 343-345, 677-682.

(6) Figure 2:The authors could justify the reason for doing the experiments before moving into the results they got.

Thanks so much for Reviewer #2 comments and efforts for our manuscript. We have revised it carefully. Please check the methods and results in the manuscript, please check them in Line 126-130, 144-147, 174-179, 213-217, 343-345, 677-682. We have added more explanation of the figure panels in the results section, and increased the correlation between methods, in order to show how to carry out the experiment logically and interpret the results.

What concentration of linalool was used?

Thanks so much for Reviewer #2 comments and efforts for our manuscript. We have revised it carefully. Please check it in Line 992-996. The mycelium treated with 6×MIC (0.3%) linalool was observed by Confocal laser scanning microscopy (CLSM), and the mycelium treated with 1×MIC 0.05% linalool was observed by Scanning Electron Microscope (SEM) and transmission electron microscopy (TEM).

The full form of DEGs has been mentioned later, but it should be mentioned in the figure legend of Figure 2 as this is the first time the term was used. Also, what is the full form of DEPs?

Thanks so much for Reviewer #2 comments and efforts for our manuscript. We have revised it carefully. Please check it in Line 168, 175, 182, 631, 998, 1001. The word DEPs in Figure 2I was incorrect, and we have changed DEPs to DEGs.

Is there a particular reason for looking into the cellular component rather than molecular function and biological processes in the GO analysis? (what I see is that Figure 2H indicates the prevalence of catalytic activity, binding, cellular, and metabolic processes as well). Also, there is not enough explanation of the observation from Figure 2I (both in the results section and figure legend).

Thanks so much for Reviewer #2 comments and efforts for our manuscript. We have revised it carefully. Please check it in Line 174-179, 998-1002 (Figure 2I). The reason we looked at cellular components rather than molecular functions and biological processes in GO analysis is because we focused more on the effects of cell membranes and cell walls. These results are closely related to and echo the results of our scanning electron microscopy (SEM) and transmission electron microscopy (TEM), and also support the results of electron microscopy. Enough explanations have added to the results and figure legend section to explain the observations from Figure 2I.

(7) Figure 3:Figures 3A and 3B: The adjusted p value is already indicated in the figures, so there is no need to add statistical significance (Asterix) to each bar. The resolution for these panels is not good and the font is too small.

Thanks so much for Reviewer #2 comments and efforts for our manuscript. We have revised it carefully. Please check the Figure 3A and 3B. We have removed statistical significance (Asterix) from Figure3A and 3B. If we are lucky, we will upload the clearest figures when the manuscript is published.

Figure 3C: the figure legend is missing (wrongly added as KEGG analysis, which should be network analysis). The numbering for the figure legends is wrong. What are the node sizes (5, 22, 40, 58) mentioned in the figure represent? Also, I wonder why ribosome biogenesis in eukaryotes has been indicated as the most enriched pathway despite its less connection to the other nodes.

Thanks so much for Reviewer #2 comments and efforts for our manuscript. We have revised it carefully. Please check the Figure 3C. Figure 3C is KEGG analysis generated by software, not network analysis. For the convenience of readers, we have made a new Figure of KEGG analysis.

Figure 3D: KEGG enrichment and GO analysis: global/local search? Which database was used as a reference?

Thanks so much for Reviewer #2 comments and efforts for our manuscript. We have revised it carefully. Please check the 633-635. Functional enrichment analysis was performed using the Kyoto Encyclopedia of Genes and Genomes (KEGG) database. KEGG pathway analysis was conducted using Goatools.

Figure 3E: why were the RNA pol structures compared? The authors did not mention anything about this panel in their results.

Thanks so much for Reviewer #2 comments and efforts for our manuscript. We have revised it carefully. Please check the line 207. We found that many DEGs related to ribosome biogenesis (Figure 3D) and RNA polymerase (Figure 3E) are down expressed. Because RNA polymerase is closely related to ribosome biogenesis, the downregulation of RNA polymerase directly affects the synthesis of ribosome-related RNAs, including rRNA, mRNA, and tRNA, thereby inhibiting ribosome production. This relationship is particularly significant in cell growth, division, and the response to external environmental changes.

Figures 3F and 3G: please mention which model is illustrated (ribbon/sphere model).

Thanks so much for Reviewer #2 comments and efforts for our manuscript. We have revised it carefully. Please check the line 1010-1015. The tertiary structure of NOP1 was displayed using a cartoon representation. Molecular docking of linalool with NOP1 was performed by enlarging the regions binding to the NOP1 activation pocket to showcase the detailed amino acid structures, which were presented using a surface model, while the small molecule was displayed with a ball-and-stick representation.

Figure 3H: this panel needs more explanation. Why were some of the ABC transporters upregulated while some were downregulated?

Thanks so much for Reviewer #2 comments and efforts for our manuscript. We have revised it carefully. It is a common phenomenon that microorganisms adjust the expression of genes related to substance transport in response to different environmental stimuli to optimize their survival strategies. The expression of ATP-binding cassette (ABC) transporters can be upregulated or downregulated due to various factors, such as environmental stimuli, metabolic demands, energy consumption, species specificity, and signaling molecules. This explains why some ABC transporters are upregulated while others are downregulated.

(8) Figure 4:There was no statistical significance shown in the figures (D-F) which makes me wonder how they worked out that there was any significant increase/decrease, as mentioned in the text. What are the p values? What is the number of replicates? What concentration of linalool was used?

Thanks so much for Reviewer #2 comments and efforts for our manuscript. We have revised it carefully. Please check the Figure 4D-F. In this study, 4 groups were established: (1) Positive control (PC) group (10 fish infected with *S. parasitica*). (2) Linalool therapeutic (LT) group (10 fish infected with *S. parasitica*, soaked in 0.00039% linalool in a 20L tank for 7 days). (3) Linalool prophylactic (LP) group (10 uninfected fish soaked in 0.00039% linalool in a 20L tank for 2 days, followed by the addition of 1×10^6^ spores/mL secondary zoospores). (4) Negative control (NC) group (10 uninfected fish without linalool treatment). Each group had 3 replicate tanks. In each group, 8 fish were utilized for immunological assays, and on day 7, blood samples were collected from the tail veins using heparinized syringes and left to coagulate overnight at 4°C. Kits from Nanjing Jiancheng Institute (Nanjing, China) were used to measure lysozyme (LZY) activity, superoxide dismutase (SOD) activity, and alkaline phosphatase (AKP) activity.

(9) Figure 5:Again, the resolution and font size are off. Please mention the full forms of the terms used in the figure legend. The interpretation of the in vivo protective mechanism of linalool is completely based on GO enrichment and KEGG pathway analysis (also some transcriptional analysis). The only wet lab validation done was by checking the mRNA level of some cytokines but that does not necessarily validate what the authors claim.

Thank you for your suggestion. We have revised it carefully. Please check all the figures and figure legend. We have increased the resolution and size of all the figures and used the full forms of the terms in figure legend. If we are lucky, we will upload the clearest figures when the manuscript is published. Currently, in the field of aquaculture research, mRNA quantification at the genetic level faces numerous challenges compared to model organisms like mice and zebra fish, primarily due to the lack of available antibodies. For instance, antibodies related to grass carp have not yet been commercialized, making protein-level studies and validations significantly more difficult. This lack of antibodies limits the progress of protein verification. However, we hope to design more experiments and validation tests in the future to gradually overcome these technical bottlenecks and provide stronger support for research in the future.

(10) Figure 6:There is not enough explanation on why and how the experiments were done. It seems like the authors already presumed that the readers know the experiments. The interpretation of the PCA plot is not clear. Why are the quadrant sizes different? How was the heat map plotted? Also, the claim of linalool regulating the gut microbiota is only dependent on the correlation analysis and there is no wet lab validation for this. The data represented in this figure is not enough to prove their hypothesis and needs further investigation.

Thanks so much for Reviewer #2 comments and efforts for our manuscript. We have revised it carefully. Please check the Figure 6. We will improve the labeling of the figure panels, provide detailed explanations of the experimental methods, including their rationale and interpretation, and clarify the connections between the methods.

The goal of PCoA is to preserve the distance relationships between samples as much as possible through the principal coordinates, thereby revealing the differences or patterns in microbial composition among different groups. For example, in our study, PCoA analysis demonstrated that the microbial compositions of the positive control (PC), linalool prophylactic (LP), and linalool therapeutic (LT) groups showed significant differences in the reduced dimensional space, possibly indicating that these treatments had a notable impact on the microbial community.

In our study, the heatmap was generated using the Majorbio Cloud Platform. This platform visualized the preprocessed microbial community data, providing an intuitive representation of the differences in microbial composition and relative abundance among samples. The platform automatically performed steps such as data normalization, color mapping, and clustering analysis, offering convenience for data analysis and interpretation.

Previous researches have shown that antibiotics and other drugs can cause alterations in gut microbiota. Therefore, we plan to study the effects of antibiotics on gut microbiota. To conduct this research, we need to isolate these microbes from the gut. Although this process is challenging, we still aim to explore the gut microbiota. If possible, we will continue to delve into interesting aspects of how antibiotics affect gut microbiota in future studies.

(11) Figure 7:This figure does not clarify how they did the interpretation. The in vivo study does not phenocopy their in vivo studies.

Thanks so much for Reviewer #2 comments and efforts for our manuscript. We have revised it carefully. we have carefully reviewed and confirmed the current experimental design and data analysis. Although we have not made any changes to Figure 7, we have further clarified the interpretation of the results in the revised manuscript, especially concerning the discrepancies between the in vivo and in vitro studies. We have added more experimental background information to help better understand the possible reasons for these differences. We hope the reviewer will understand our explanation and we look forward to your further feedback.

(12) Minor comments:Line 61: what's meant by "et al"?

Thanks so much for Reviewer #2 comments and efforts for our manuscript. We have revised it carefully. Please check it in Line 61. We have removed "et al".

Line 87-88: please add a citation referring to the earlier studies.

Thanks so much for Reviewer #2 comments and efforts for our manuscript. We have revised it carefully. Please check it in Line 109.

Line 151-152: the term "related to" has been used a couple of times. Mentioning it once in the beginning and avoiding repeating the same word might be better.

Thanks so much for Reviewer #2 comments and efforts for our manuscript. We have revised it carefully. Please check it in Line 168-171.We have rewritten this paragraph to avoid repeating the word “related to”.

How did they reconstitute the EO compounds?

Thanks so much for Reviewer #2 comments and efforts for our manuscript. We have revised it carefully. The EO compounds we used in our experiments were partially extracted from essential oils in the laboratory and partially purchased from ThermoFisher (USA).

Line 544: needs explanation of how there was a 2-fold dilution in the concentrations shown in the figure compared to the concentrations mentioned here.

Thanks so much for Reviewer #2 comments and efforts for our manuscript. We have revised it carefully. We set the concentration of MIC assay for mycelium to be 0.8%, 0.4%, 0.2%, 0.1%, 0.05%, 0.025%, 0.0125%, and 0.00625%, and the concentration of MIC assay for spores to be 0.4%, 0.2%, 0.1%. 0.05%, 0.025%, 0.0125%, 0.00625%. Figure 1B shows the MIC determination of linalool on spores, while the MIC determination of mycelium is not shown.

Line 546: remove "were".

Thanks so much for Reviewer #2 comments and efforts for our manuscript. We have revised it carefully. Please check it in Line 573. We have removed "were".

Line 555: what concentration of malachite green and tween 20 was used?

Thanks so much for Reviewer #2 comments and efforts for our manuscript. We have revised it carefully. Please check it in Line 579-580. 2.5mg /mL malachite green and 1% Tween 20 were used.